# 1-Undecene from *Pseudomonas aeruginosa* is an olfactory signal for flight-or-fight response in *Caenorhabditis elegans*

Deep Prakash[1], Akhil MS[1], Buddidhathi Radhika[2], Radhika Venkatesan[2,3], Sreekanth H Chalasani[4] & Varsha Singh[1,*]

## Abstract

Animals possess conserved mechanisms to detect pathogens and to improve survival in their presence by altering their own behavior and physiology. Here, we utilize *Caenorhabditis elegans* as a model host to ask whether bacterial volatiles constitute microbe-associated molecular patterns. Using gas chromatography–mass spectrometry, we identify six prominent volatiles released by the bacterium *Pseudomonas aeruginosa*. We show that a specific volatile, 1-undecene, activates nematode odor sensory neurons inducing both flight and fight responses in worms. Using behavioral assays, we show that worms are repelled by 1-undecene and that this aversion response is driven by the detection of this volatile through AWB odor sensory neurons. Furthermore, we find that 1-undecene odor can induce immune effectors specific to *P. aeruginosa* via AWB neurons and that brief pre-exposure of worms to the odor enhances their survival upon subsequent bacterial infection. These results show that 1-undecene derived from *P. aeruginosa* serves as a pathogen-associated molecular pattern for the induction of protective responses in *C. elegans*.

**Keywords** 1-undecene; *Caenorhabditis elegans*; flight-or-fight response; olfaction; *Pseudomonas aeruginosa*

**Subject Category** Neuroscience

**The EMBO Journal (2021) 40: e106938**

## Introduction

Threat perception is one of the key drivers of behavioral and physiological responses in living organisms. Animals use their senses of vision, taste, auditory perception, and olfaction to perceive threats and engage in flight-or-fight mechanisms to improve survival. In case of infection, the sensing of pathogens via pattern recognition initiates physiological responses in the host including immune response to enhance survival. The chemical nature of pattern varies greatly from bacterial lipopolysaccharides to peptides or even volatile compounds (Stensmyr *et al*, 2012; Bufe *et al*, 2019). Volatiles are increasingly being recognized as a mode of communication between plants in response to herbivory and other threats (Erb *et al*, 2015). However, the contribution of volatiles in driving immune responses in animals remains poorly understood. In this study, we have used bacterivorous nematode *Caenorhabditis elegans* to identify volatiles that serve as molecular pattern to alter host behavior and immune response.

*Caenorhabditis elegans* forages for food bacteria in decaying organic matter (Schulenburg & Félix, 2017). A well-developed chemosensory system including an odor sensory system enables worms to efficiently engage in food search behavior as well as avoid pathogens (Shtonda & Avery, 2006; Pradel *et al*, 2007). *C. elegans* shows chemotaxis response to a wide range of volatile molecules including alcohols, ketones, amines, aldehydes, organic acids, and aromatic and heterocyclic compounds (Bargmann *et al*, 1993), many of these are product of bacterial secondary metabolism indicating their role in sensory perception (Worthy *et al*, 2018; Foster *et al*, 2020).

*Caenorhabditis elegans* exhibits both fight and flight responses to *Pseudomonas aeruginosa*, a ubiquitous bacterium and an opportunistic human pathogen. Several studies on the interaction of *C. elegans* with *P. aeruginosa* have led to the elucidation of innate immune mechanisms that confer protection to the worms (Kim *et al*, 2002; Singh & Aballay, 2006; Estes *et al*, 2010; Irazoqui *et al*, 2010). The flight response, the ability of worms to avoid a lawn of *P. aeruginosa*, is a behavior termed "aversion" (Zhang *et al*, 2005; Styer *et al*, 2008). Anatomical changes in worm's intestine during *P. aeruginosa* infection have been linked to aversion response of worms (Singh & Aballay, 2019). This behavior has also been linked to the sensation of water-soluble secondary metabolites, pyochelin and phenazine-1-caboxamide of *P. aeruginosa*, by chemosensory neuron ASJ (Meisel *et al*, 2014). Evidence also suggests that odor

1  Department of Molecular Reproduction, Development and Genetics, Indian Institute of Science, Bangalore, India
2  National Center of Biological Sciences, Bangalore, India
3  Department of Biological Sciences, Indian Institute of Science Education and Research, Mohanpur, India
4  Salk Institute for Biological Studies, La Jolla, CA, USA
   *Corresponding author. Tel: +91 80 22933464; E-mail: varsha@iisc.ac.in

sensory neurons regulate chemosensation and aversive olfactory learning during *P. aeruginosa* infection (Zhang *et al,* 2005) suggesting a possible involvement of volatile metabolites of *P. aeruginosa* in *C. elegans* response. The innate ability of worms to respond to the olfactory cues from *P. aeruginosa* was shown recently (Ooi & Prahlad, 2017). Additionally, genes involved in nitric oxide production in *P. aeruginosa* have been implicated in eliciting aversion in worms (Hao *et al,* 2018). However, the nature of *P. aeruginosa* volatile molecules which induce immune response by serving as microbe-associated molecular pattern in *C. elegans* remains unclear.

In this study, we asked whether bacterial volatiles serve as molecular patterns. We utilized odor sensory mutants of *C. elegans* and specific volatiles from bacterium *P. aeruginosa* to study aversion response and immune response in the host. By analyzing volatile organic compounds produced by the bacterium, we identify 1-undecene, an 11-carbon olefin, as the aversive volatile signal. The volatile induces aversion response in *C. elegans* and calcium signaling in AWB odor sensory neurons. Finally, we show that 1-undecene serves as a molecular pattern and induces upregulation of a subset of immune response genes specific to *P. aeruginosa* in worms, in AWB neuron-dependent manner.

# Results

## Odor sensation controls *C. elegans* flight response to *P. aeruginosa*

In a systematic study of aversion response in *C. elegans,* we exposed adult worms to an old lawn of *P. aeruginosa* PA14 and scored lawn occupancy as a measure of aversion response every 4 h. The lawn occupancy of naive worms on the *P. aeruginosa* lawn reduced from 100% at 0 h to ~ 20% at 12 h of exposure indicating an increase in aversion with time (Fig 1A and B, Movie EV1A). To understand whether odor sensation in worms contributed to lawn leaving, we examined the aversion response of *odr-3(n2150)* and *odr-3(n2046)* olfaction defective mutants. ODR-3 encodes a G protein alpha subunit required for normal chemotaxis to odorants (Bargmann *et al,* 1993; Roayaie *et al,* 1998). We found that *odr-3* mutants were defective in aversion response on *P. aeruginosa* lawn with high (60–90%) lawn occupancy even at 12 h of exposure (Fig 1A and B, Movie EV1B). Severely muted aversion response in *odr-3* mutant worms suggested that volatile cues from the *P. aeruginosa* lawn contribute to aversion response.

To understand whether olfaction-dependent aversion response is protective, we analyzed the survival of N2 and *odr-3* mutant worms on a partial lawn (aversion allowed) and full lawn (aversion not allowed) of *P. aeruginosa*. While *odr-3* mutants were more susceptible to infection than N2 on the partial lawn, N2 and *odr-3* mutant worms had comparable survivals on the complete lawn of PA14 (Fig 1C–E), suggesting that olfaction-dependent aversion response of worms enhances their survival. In a volatile-driven choice assay between 8-h (young) and 24-h (old) lawn of *P. aeruginosa,* in a tripartite plate, we observed that N2 worms preferred the younger lawn over the older lawn, but this preference was absent in *odr-3 (n2150)* mutant (Fig 1F). This suggested that the older lawn of *P. aeruginosa* might produce volatile chemorepellents. It was shown recently that bloating in *C. elegans* gut upon long exposure of

worms to *P. aeruginosa* leads to aversion response (Singh & Aballay, 2019). However, we found no bloating in N2 worms upon 12 h of exposure and the gut lumen width in *E. coli* and *P. aeruginosa* exposed worms were comparable (Fig 1G and H), suggesting that olfaction-mediated aversion response seen at 12 h exposure is not linked to bloating in worms. Taken together, these results suggested that the volatiles produced by *P. aeruginosa* lawn induce a protective aversion response in olfaction-competent *C. elegans*.

## 1-Undecene from *P. aeruginosa* elicits aversive response in *C. elegans*

To identify the aversion-inducing volatile(s) of *P. aeruginosa*, we performed solid-phase microextraction (SPME) and gas chromatography–mass spectrometry (GC-MS) analysis of volatiles. Volatiles were collected directly from the headspace between two plates containing *P. aeruginosa* lawns (schematic in Appendix Fig S1A). *P. aeruginosa* had six abundant volatiles, dimethyl sulfide (DMS), dimethyl disulfide (DMDS), pyrrole, 1,4-dichlorobenzene, hexanoic acid, 2-ethyl-, methyl ester, and 1-undecene (Fig 2A) that were absent in the media control and *E. coli* OP50 headspace (Appendix Fig S1B and C).

We examined the chemotaxis of N2 worms to individual volatiles and found that only 1-undecene was an aversive signal for worms (Fig 2B, Movie EV2A). Using a two-plate arrangement, to ensure delivery of only volatile form of 1-undecene, we confirmed that worms show aversion to 1-undecene odor (Appendix Fig S2A). Most animals deploy various escape strategies upon sensing threat signals, *C. elegans* exhibits enhanced roaming and omega turns upon encountering aversive signals (Liu *et al,* 2018). To test whether 1-undecene is indeed an aversive signal, we examined the locomotion of worms in presence of 1-undecene. Worms executed omega bend in response to 1-undecene in 16 out of 17 worms (Fig 2C, Movie EV3) consistent with the response to an aversive signal. Finally, we also examined *undA* mutant of PA14 (PA14_53120 locus), incapable of producing 1-undecene. UndA is a non-heme iron oxidase necessary for the biosynthesis of 1-undecene in different *P. aeruginosa* strains (Rui *et al,* 2014). As expected, transposon insertion mutant of *undA* did not produce any 1-undecene in GC-MS analysis (Appendix Fig S2C). We performed a choice assay between odors of PA14 and *undA* lawn and found that worms preferred *undA* over PA14 (Fig 2D). We also examined the transcript levels of *undA* in 8-h old and 24-h old lawn of bacteria and found that *undA* expression was higher in the older lawn (Appendix Fig S2D). We also observed enhanced roaming by worms in an arena exposed to 1-undecene odor compared to the arena without the repellent (Fig 2E and Appendix Fig S2B). In all, our analysis of volatiles from the headspace of aversion-inducing *P. aeruginosa* lawns revealed that 1-undecene produced by *P. aeruginosa* is an aversive signal for the host.

## 1-Undecene is sensed by AWB odor sensory neurons of *C. elegans*

To test whether olfaction in worms is necessary for sensing 1-undecene, we tested the chemotaxis response of various olfaction-compromised mutants. To begin with, we examined the chemotaxis of N2 and *odr-3* mutant worms to 1-undecene. We found that the sensing of 1-undecene was dependent on ODR-3 (Fig 3A, Movie

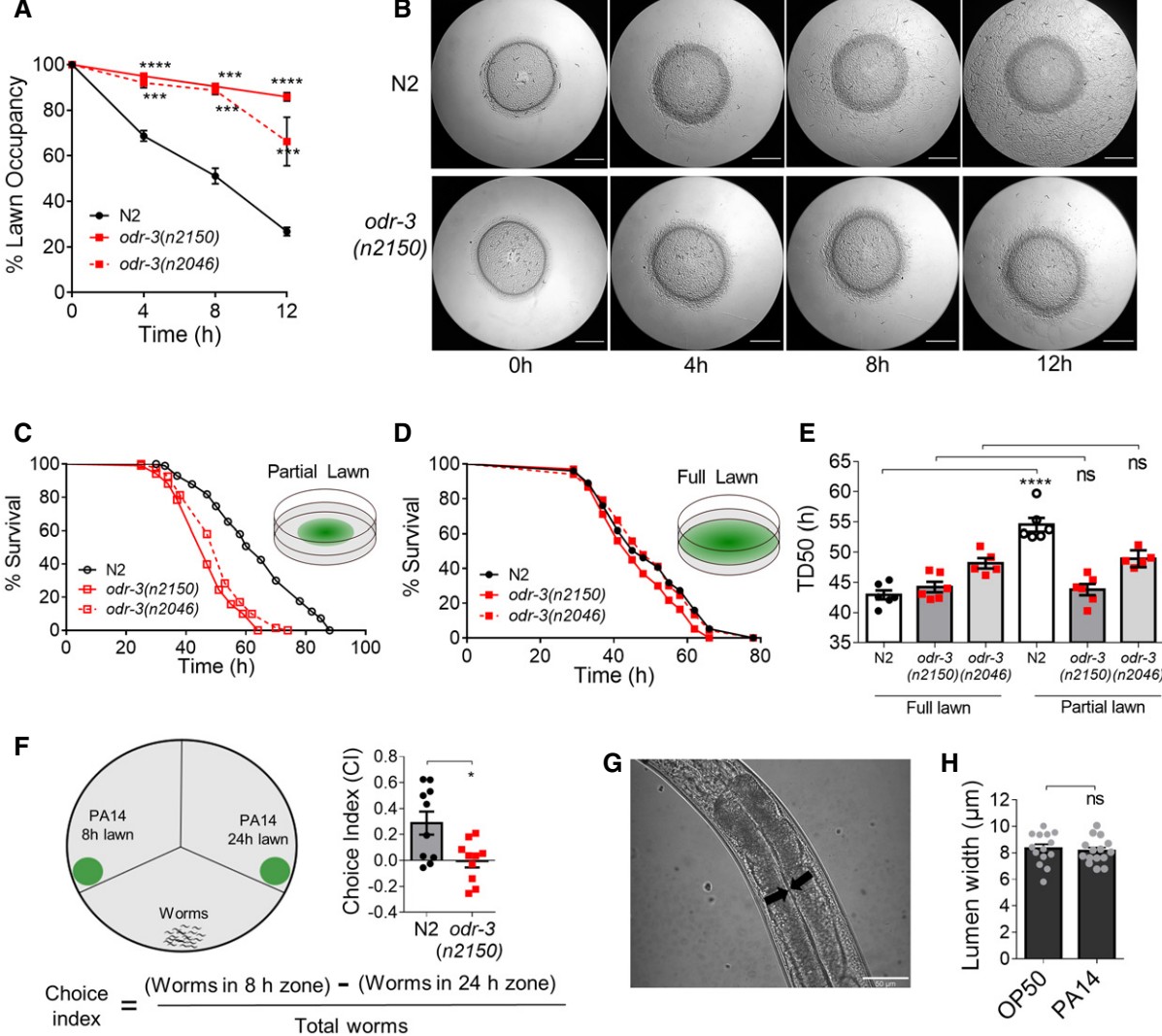

**Figure 1. Odor sensation controls *C. elegans* flight response to *P. aeruginosa*.**

A  Time course of avoidance of *P. aeruginosa* lawn by N2, *odr-3(n2150)*, and *odr-3(n2046)* worms. $n \geq 3$ assays. ***$P \leq 0.001$, ****$P \leq 0.0001$ as determined by two-tailed unpaired *t*-test. Error bars indicate SEM.

B  Images of *P. aeruginosa* lawn at different time intervals showing avoidance response of N2 and *odr-3(n2150)* mutant worms. Scale bar = 5 mm.

C  Kaplan–Meier survival curve of N2, *odr-3(n2150)*, and *odr-3(n2046)* worms on partial lawn of *P. aeruginosa* at 25°C.

D  Kaplan–Meier survival curve of N2, *odr-3(n2150)*, and *odr-3(n2046)* worms on full lawn of *P. aeruginosa* at 25°C.

E  Time required for 50% of worms to die (TD$_{50}$) on partial lawn and full lawn of *P. aeruginosa*. Individual data points indicate replicates with ~ 100 worms each. ****$P \leq 0.0001$, ns (not significant) $P > 0.05$ as determined by two-tailed unpaired *t*-test. $n \geq 3$ assays. Error bars indicate SEM.

F  Schematic of young (8 h) and old (24 h) lawn choice assay in a tripartite plate and choice index. $n \geq 3$ assays. Individual data points indicate replicates with ~ 40 worms each. *$P \leq 0.05$ as determined by two-tailed unpaired *t*-test. Error bars indicate SEM.

G  Representative image of intestinal lumen width, indicated by arrows, in N2 worm exposed to *P. aeruginosa* for 12 h. Scale bar = 50 μm.

H  Quantification for the width of intestinal lumen in N2 exposed to *E. coli* and *P. aeruginosa* for 12 h. ns (not significant) $P > 0.05$ as determined by two-tailed unpaired *t*-test. $n = 3$. Error bars indicate SEM.

EV2B). Further, to identify olfactory neurons necessary for sensing, we used mutants or ablation lines for odor sensory neurons AWA, AWB, and AWC of worms. We first examined *lim-4*(ky403) and *lim-4*(yz12) worms that lack functional AWB neurons (Sagasti *et al*, 1999) and found that they showed no response to 1-undecene. However, AWC-ablation worms (Beverly *et al*, 2011) and *odr-7*(ky4) worms lacking functional AWA neurons (Sengupta *et al*, 1994),

along with AWA-ablation worms, had a normal aversion response to 1-undecene, comparable to N2 worms. ODR-3 is also expressed in non-olfactory neurons such as nociceptive neuron ASH which can respond to odors (Yoshida *et al*, 2012). We also used ablation for ASH neurons to show that this neuron is not involved in response to 1-undecene. (Fig 3B). These results suggested that the sensing of 1-undecene is dependent on functional AWB neurons.

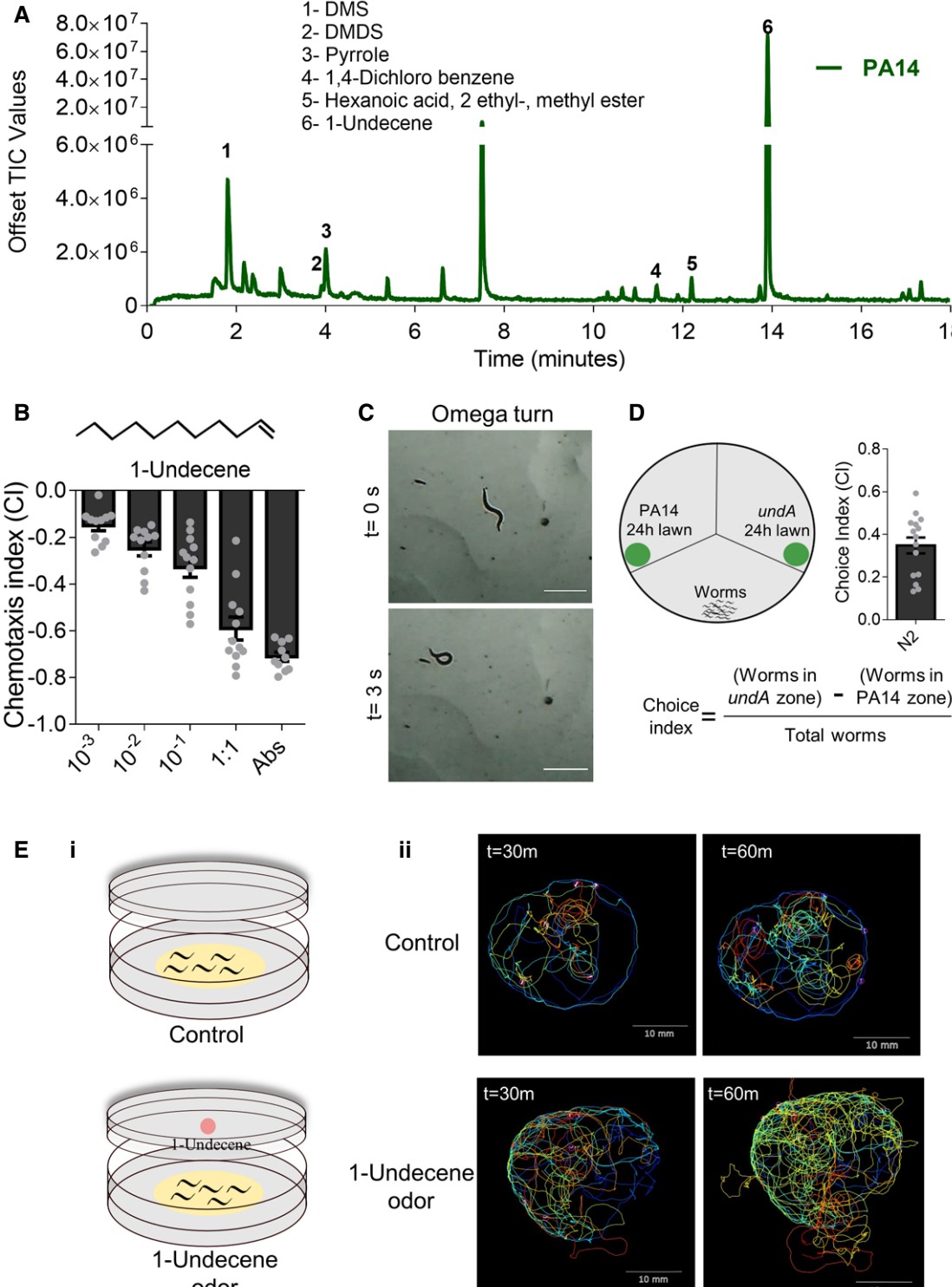

**Figure 2. 1-Undecene from *P. aeruginosa* elicits aversive response in *C. elegans*.**

A   GC-MS/MS profile of volatiles produced by 24-h old lawn of *P. aeruginosa* PA14.

B   Chemotactic response of N2 worms to varying dilutions of 1-undecene. $n \geq 3$ assays. Error bars indicate SEM.

C   Omega turn in N2 worms under 1-undecene exposure. Scale bar = 1 mm.

D   Schematic of PA14 (24 h) and *undA* mutant (24 h) bacterial lawn choice assay in a tripartite plate and choice index. $n \geq 3$ assays. Error bars indicate SEM.

E   (i) Schematic of setup used to record the movement of 5 worms each under control and 1-undecene odor exposure. (ii) Movement trajectories of N2 worms on *E. coli* OP50 lawn exposed to 1-undecene odor for 30 min and 60 min, each color represents the trajectory of single worm. Scale bar = 10 mm.

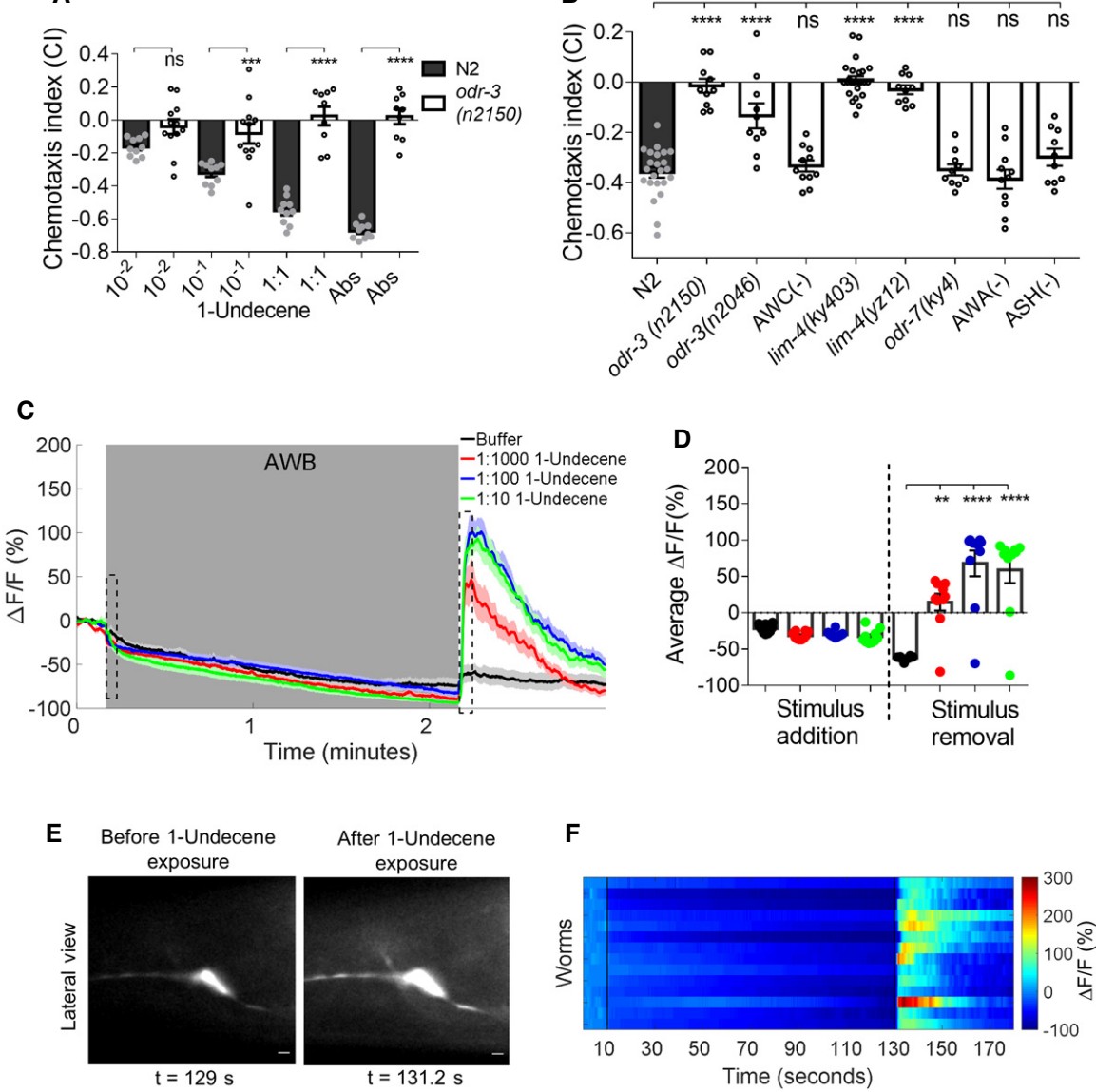

**Figure 3. 1-Undecene is sensed by AWB odor sensory neurons of *C. elegans*.**

A   Chemotactic dose response of N2 and *odr-3(n2150)* worms to 1-undecene. $n \geq 3$ assays. ns (not significant) $P > 0.05$, ***$P \leq 0.001$, ****$P \leq 0.0001$ as determined by one-way ANOVA, followed by Dunnett's multiple comparison test. Error bars indicate SEM.

B   Chemotaxis of N2, *odr-3(n2150)*, *odr-3(n2046)*, AWC(-), *lim-4(ky403)*, *lim-4(yz12)*, *odr-7(ky4)*, AWA(−), and ASH(−) worms to 1-undecene at 1:10 dilution. $n \geq 3$ assays. ns (not significant) $P > 0.05$, ****$P \leq 0.0001$ as determined by one-way ANOVA, followed by Dunnett's multiple comparison test. Error bars indicate SEM.

C   Average calcium responses of AWB::GCaMP3 worms exposed to 1-undecene at indicated concentrations. Each worm was recorded for 180 s. Worms were under stimulus between 11–130 s, window shown in gray, and stimulus withdrawal beyond 130 s. Shaded regions around the curves represent SEM. $n = 3$. Two dashed box represents 10 s time window after stimulus addition and stimulus removal, respectively.

D   Average percentage change in $\Delta F/F$ for 10 s time window after stimulus addition and withdrawal (dashed boxes in C), each data point represents average data of $\geq 13$ worms. **$P \leq 0.01$, ****$P \leq 0.0001$ as determined by one-way ANOVA, followed by Dunnett's multiple comparison test. Error bars indicate SEM.

E   Representative image of GCaMP3 fluorescence in AWB neuron during 1-undecene exposure (129 s on *x*-axis of Fig 3C) and after 1-undecene withdrawal (131.2 s on *x*-axis of Fig 3C). Scale bar = 10 μm.

F   Heat map of ratio $\Delta F/F$ (change in fluorescence to total fluorescence) in AWB::GCaMP3 neuron upon the addition (10 s) or the removal (130 s) of 1-undecene. Each row represents an individual worm recorded for 180 s under 1:100 dilution of 1-undecene, in 11–130 s window.

To further confirm the role of AWB neurons in sensing 1-undecene, we examined the odor-evoked calcium response of AWB neurons expressing genetically encoded calcium sensor GCaMP (Chalasani *et al*, 2007). In AWB neurons, we observed a dose-dependent, stimulus-withdrawal response within 1–2 s of removal of stimulus from the nose of the worm (Fig 3C and D), consistent

with the requirement of AWB neurons in chemotaxis response (Fig 3B). Withdrawal-induced calcium response in AWB neurons has been reported for other aversive volatiles (Ha *et al*, 2010). Stimulus-withdrawal response was observed in AWB neurons in all the worms examined (Fig 3E and F). We observed none or very small calcium response to 1-undecene withdrawal in AWA, AWC^on, or AWC^off neurons (Appendix Fig S3) consistent with their dispensability in chemotaxis response. These results confirmed that AWB neurons are necessary for sensing and responding to 1-undecene.

### 1-Undecene odor induces pathogen-specific immune response in *C. elegans*

To understand whether 1-undecene serves as a microbe-associated molecular pattern for *C. elegans*, we examined the activation of immune response in worms in the presence of only the volatile signals. We reasoned that sensing of *P. aeruginosa*-associated molecular patterns would be integrated into mechanisms to activate innate immune responses specific to this pathogen. To test this, we analyzed the expression of *P. aeruginosa*-specific immune response genes of *irg* family which are expressed under the control of ZIP-2 transcription factor (Estes *et al*, 2010) or immune effector genes under the control of p38 MAP kinase and other immune response genes (Troemel *et al*, 2006). Interestingly, we found that worms pre-exposed to 1-undecene for just two hours had 2- to 30-fold induction of *irg-1, irg2*, and *irg-3* genes compared to the naive worms (Fig 4A). Transcripts for *irg-7, dod-24, mul-1, clec-67, C17H12.8* were not upregulated while F01D5.5 was downregulated in 1-undecene-exposed worms (Fig 4A) suggesting that not all but one specific immune response pathway, regulated by ZIP-2, is engaged upon 1-undecene sensing. Next, we examined the expression of transcripts in *zip-2(tm4248)* mutant and found that upregulation of *irg-1* and *irg-2* induced by 1-undecene was significantly lower than in N2 worms (Fig 4B). Using an *irg-1p*::GFP reporter, we confirmed increased reporter expression in 1-undecene-exposed worms compared to naive worms (Fig 4C). Next, we examined transcript levels in *odr-3* and *lim-4* mutants and found that the induction of *irg-1, irg-2,* and *irg-3* transcripts during 1-undecene odor exposure was dependent on ODR-3 and it also required functional AWB neurons (Fig 4D and E) confirming the involvement of 1-undecene-responsive AWB neurons in mounting the immune response.

*odr-3* and *lim-4* mutants did not regulate the expression of other *P. aeruginosa*-specific immune response genes suggesting the specificity of 1-undecene response towards ZIP-2-regulated immune response (Appendix Fig S4A and B). To understand whether odor-induced upregulation of ZIP-2 pathway protects worms from infection, we pre-exposed worms to 1-undecene followed by chronic infection with *P. aeruginosa*. As shown in Fig 4F, worms pre-exposed to 1-undecene had significantly better survival than naive worms (also see Fig 4G). To further confirm that *irg* upregulation is induced by 1-undecene odor emanating from *P. aeruginosa* lawns, we examined the levels of transcripts in worms exposed to PA14 odor or *undA* mutant odor. We found that the induction of *irg-1, irg-2,* and *irg-3* by PA14 odor was completely abrogated in worms exposed to *undA* odor (Fig 4H). Moreover, the survival of worms on *undA* was significantly lower than on wild-type *P. aeruginosa* PA14 (Appendix Fig S4E and F) suggesting that 1-undecene-induced immune response contributes to the survival of worms during *P. aeruginosa* infection. These results suggested that 1-undecene present in PA14 odor is necessary and sufficient to induce immune effector upregulation.

To confirm that 1-undecene-induced response was specific to *P. aeruginosa*, we also examined the expression of eight immune response genes induced 10- to 100-fold in worms in response to Gram-positive bacteria, *Enterococcus faecalis* and *Staphylococcus aureus,* or pathogenic yeast *Cryptococcus neoformans* (Dasgupta *et al*, 2020). We found that 1-undecene odor exposure showed little or no induction in expression of transcripts of *fmo-2, acs-2, lipl-1, lipl-3, cpr-4, cpr-5, asp-14,* and *lys-3* (Appendix Fig S4D). The inability of 1-undecene to induce *fmo-2* could also be observed using a *fmo-2p*::GFP reporter (Appendix Fig S4C). We also analyzed the induction of heat shock response or oxidative stress response using *hsp-16.2p*::GFP and *gst-4p*::GFP, respectively (Appendix Fig S4G and H) (Link *et al*, 1999; Link & Johnson, 2002). We examined two additional markers, ATF-4 and HSP-4 (Shen *et al*, 2001; Glover-Cutter *et al*, 2013), to test the effect of 1-undecene exposure on homeostasis and found no appreciable change (Appendix Fig S4I). Thus, we found that 1-undecene odor exposure did not disrupt cellular homeostasis.

Based on our study, we propose a model for volatile-based pattern recognition in *C. elegans*. We show that 1-undecene, a volatile produced abundantly by *P. aeruginosa*, is sensed by AWB odor

---

**Figure 4.  1-Undecene odor induces pathogen-specific immune response in *C. elegans*.**

A  Real-time PCR analysis of *P. aeruginosa*-specific immune response genes in naive and 1-undecene odor-exposed N2 worms. *n* = 3. *$P \leq 0.05$, **$P \leq 0.01$ as determined by two-tailed unpaired *t*-test. Error bars indicate SEM. The negative values are arrived at by representing FC value less than 1 as ($-1/$FC).

B  Real-time PCR analysis of *irg-1, irg-2,* and *irg-3* genes in N2 and *zip-2(tm4248)* worms exposed to 1-undecene odor upon respective naive worms. *n* = 3. ns (not significant) $P > 0.05$, *$P \leq 0.05$, **$P \leq 0.01$ as determined by two-tailed unpaired *t*-test. Error bars indicate SEM.

C  *irg-1p*::GFP induction in naive worms and worms exposed to *P. aeruginosa* (6 h) and 1-undecene odor (2 h). Scale bar = 500 μm.

D  Real-time PCR analysis of *irg-1, irg-2,* and *irg-3* genes in N2, *odr-3(n2150),* and *odr-3(n2046)* worms exposed to 1-undecene odor upon respective naive worms. *n* = 3. *$P \leq 0.05$, **$P \leq 0.01$ as determined by two-tailed unpaired *t*-test. Error bars indicate SEM.

E  Real-time PCR analysis of *irg-1, irg-2,* and *irg-3* genes in N2, *lim-4(ky403),* and *lim-4(yz12)* worms exposed to 1-undecene odor upon respective naive worms. *n* = 3. *$P \leq 0.05$, **$P \leq 0.01$, ***$P \leq 0.001$ as determined by two-tailed unpaired *t*-test. Error bars indicate SEM.

F  Kaplan–Meier survival curve on the full lawn of *P. aeruginosa* for N2 (naive) worms and N2 worms pre-exposed to 1-undecene odor. Survival assay was performed at 20°C.

G  Time required for 50% of worms to die (TD$_{50}$) on lawn of *P. aeruginosa*. Each data point indicates replicates with ~ 100 worms each. *n* = 3 assays. **$P \leq 0.01$ as determined by two-tailed unpaired *t*-test. Error bars indicate SEM.

H  Real-time PCR analysis of *irg-1, irg-2,* and *irg-3* genes in N2 worms exposed to odor of *undA* mutant upon PA14. *n* = 3. *$P \leq 0.05$, **$P \leq 0.01$ as determined by two-tailed unpaired *t*-test. Error bars indicate SEM.

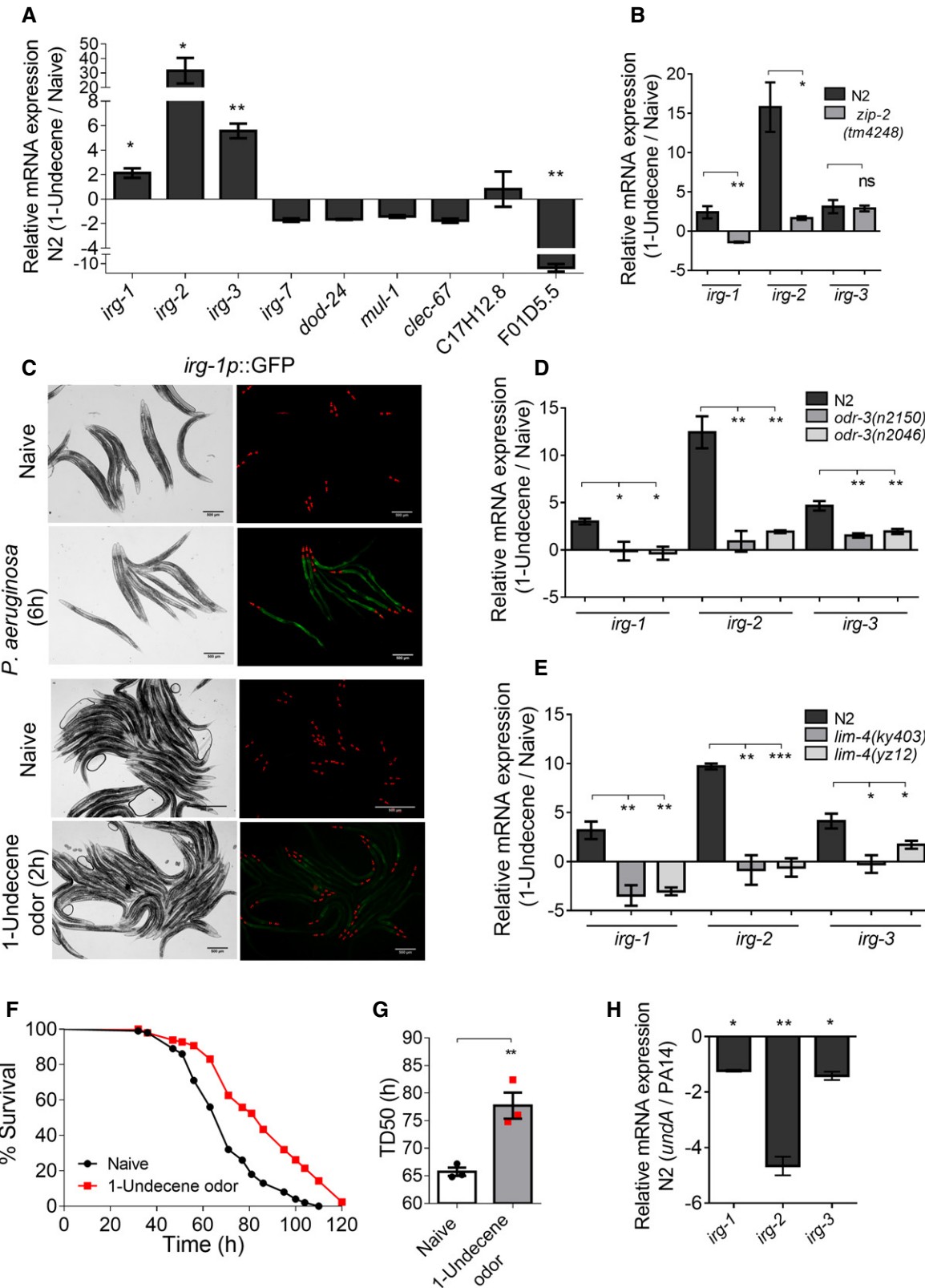

**Figure 4.**

sensory neurons of *C. elegans* leading to activation of behavioral response and immune response specific to the pathogen leading to enhanced survival of the host (Fig 5).

## Discussion

Our study provides molecular and neuronal bases of the flight and fight responses of *C. elegans* upon exposure to pathogenic bacterium *P. aeruginosa*. We show that a specific volatile organic compound, 1-undecene, produced by the pathogen *P. aeruginosa* is sensed as an aversive signal by *C. elegans*. We show that olfaction-mediated modulation of behavior by 1-undecene is dependent on neuronal signaling in the AWB odor sensory neurons. Not only does 1-undecene initiate protective aversion responses in individual worms, but it also induces *P. aeruginosa*-specific immune response and enhances survival during infection providing evidence that 1-undecene is a pathogen-associated molecular pattern (PAMP) for worms.

To qualify 1-undecene as a molecular pattern, we first asked whether 1-undecene is a *Pseudomonas*-specific signal? The presence of 1-undecene in *P. aeruginosa* headspace has been reported earlier and has been suggested as a *Pseudomonas*-specific volatile (Goeminne *et al*, 2012; Rui *et al*, 2014). Previous studies have reported that all *Pseudomonas* species have *undA* gene (Rui *et al*, 2014). Moreover, we also could not detect 1-undecene in *E. coli* OP50, *Salmonella typhimurium,* and *Enterococcus faecalis* headspace or find it in the published literature (Siripatrawan, 2008; Zhu *et al*, 2010; Worthy *et al*, 2018). This provided additional evidence that 1-undecene is a signature for *Pseudomonas* species and perceived by worms as a molecular pattern. 1-Undecene produced by *Pseudomonas* species is also known to affect fungus *Phytophthora infestans* and plants (Hunziker *et al*, 2015; Lo Cantore *et al*, 2015), suggesting that this volatile mediates inter-kingdom interactions.

The most relevant evidence for 1-undecene as a pathogen-associated molecular pattern comes from the fact that exposure of worms to volatile alone induces upregulation of immune response genes specific to *P. aeruginosa*. Specifically, ZIP-2 transcription factor-regulated genes, *irg-1*, *irg-2,* and *irg-3,* are induced in an AWB neuron-dependent manner. ZIP-2 transcription factor is required for upregulation of *irg-1* and *irg-2* in the intestine during 1-undecene odor exposure providing evidence for a brain-gut axis of immunity. Moreover, 1-undecene volatile failed to induce each one of the 8 effectors specific to pathogenic Gram-positive bacteria and yeast we tested. We also did not find evidence for induction of heat shock response or oxidative stress response. All this evidence suggests that 1-undecene is a *P. aeruginosa*-specific PAMP for *C. elegans*.

Chemosensation-based perception of pathogens is significant for worms in light of the fact that worms do not have traditional pattern recognition receptors of Toll and Nod family of pattern recognition receptor, except TOL-1 which has a limited role in *C. elegans* microbe interactions (Pradel *et al*, 2007; Tenor & Aballay, 2008; Brandt & Ringstad, 2015). However, worms have a large sensory repertoire of ~ 1,500 putative G protein-coupled receptors in the genome (Robertson, 1998), which likely serve the nematode well in a complex sensory environment created by a diverse range of microbes present in rotting vegetation, the natural habitat of *C. elegans*. A number of stimuli sensed by *C. elegans* (Bargmann *et al*, 1993) are bacterial secondary metabolites suggesting that many of the present-day sensory mechanisms may have evolved and been retained in this bacterivore to influence its ability to find food and to prevent infection.

At least two lines of evidence for perception of bacterial secondary metabolites by worms, allowing them to avoid bacteria, exist. Serrawettin W2, a surfactant produced by *S. marcescens,* is a non-olfactory cue that contributes to the aversion response of *C. elegans* to this bacterium (Pradel *et al*, 2007). Pyochelin and phenazine-1-carboxamide produced by *P. aeruginosa* are examples

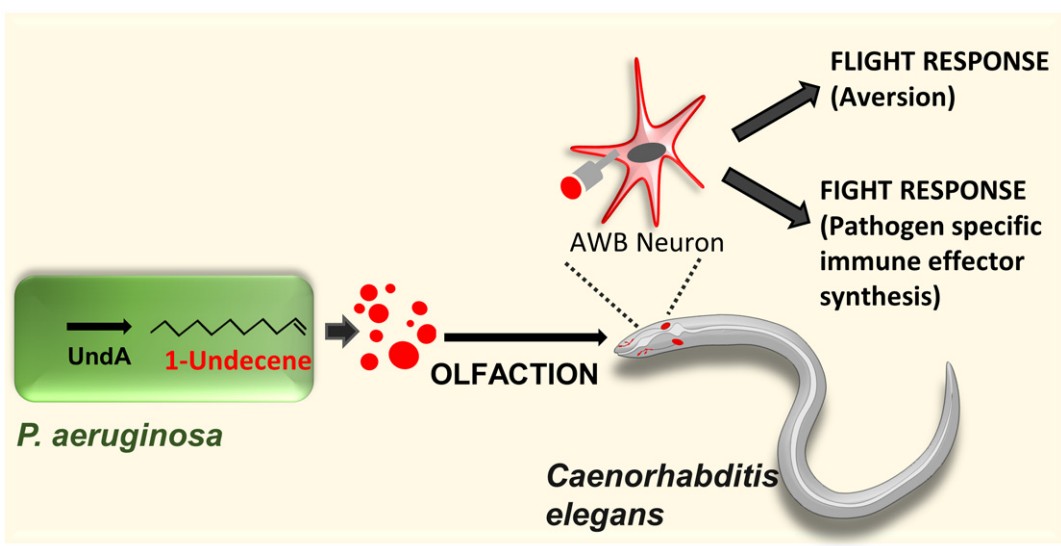

**Figure 5.  Model for olfaction driven flight and fight response in** *C. elegans*.

A non-heme oxidase UndA is required for the production of a 11-carbon olefin, 1-undecene, a volatile. This volatile induces calcium signaling in AWB odor sensory neurons leading to aversion response (FLIGHT) as well as upregulation of pathogen-specific immune response genes (FIGHT) and resistance in *C. elegans*.

of non-volatile, water-soluble molecules sensed by *C. elegans* leading to heterotrimeric G-protein signaling in ASJ chemosensory neurons (Meisel *et al*, 2014). It is important to ask what is the contribution of olfaction versus taste in the perception of *P. aeruginosa* by *C. elegans*? Our study suggests that ODR-3-based olfaction contributes to aversion response in a large fraction of population of worms suggesting that 1-undecene is a physiologically relevant stimulus for worms.

In nature, worms have to move from a piece of rotting fruit or vegetation to another at considerable distance. Under these circumstances, olfaction might allow worms to sense the aversive signals (or attractant) quicker than using the contact-based perception of taste or texture. This ability can allow worms to make appropriate decisions without incurring contact-mediated damage due to toxins. Our study shows that a unique volatile organic compound released by pathogen serves as a PAMP. *C. elegans* can use such cues to engage in flight-or-fight responses to increase its chances of survival. Additional studies of microbe–host interactions would reveal the extent of usage of olfaction in the animal kingdom for perception of pathogens.

# Materials and Methods

### Strains and growth media

All *C. elegans* strains were maintained as hermaphrodite at 20°C, on nematode growth media (NGM) plates seeded with *E. coli* OP50, as previously described (Brenner, 1974). The N2 (Bristol) was used as *C. elegans* wild-type strain. All strains used in this study are listed in Appendix Table S1. All experiments done with *C. elegans* were conducted at 25°C unless otherwise mentioned.

### Aversion response assay

All the aversion assays were performed on modified NGM agar plates called slow killing (SK) agar plates used for *P. aeruginosa* infection study. 3 ml of bacterial culture was grown at 37°C for 12 h. The assay plates were spot dried with 50 μl of overnight grown bacterial culture and incubated for 12 h at 37°C. Approximately 40–50 young adult worms were introduced in the center of the bacterial lawn and the plates were incubated at 25°C. The lawn occupancy of the worms was scored at different time intervals (4 h, 8 h, 12 h) on the basis of their position in the lawn, and the data were used to plot the lawn occupancy of worms with time. Each data point in the data set represents one assay plate with 40–50 worms each. The percent of lawn occupancy was calculated by the formula: *(Number of worms inside the lawn/Total number of worms)*.

### Chemotaxis assays

All chemotaxis assays were performed in 90-mm buffered agar plate (1 mM $CaCl_2$, 1 mM $MgSO_4$, 25 mM $KPO_4$ buffer and 2% agar). The chemicals used in this study were diluted with chloroform (solvent control). 2 μl of test or solvent was spotted on opposite sides of the assay plate (Fig S1D). Young adult worms were washed twice with S-basal buffer and once with assay buffer (1 mM $CaCl_2$, 1 mM $MgSO_4$, 25 mM $KPO_4$ buffer). Approximately 60–80 worms were

placed in the center of the assay plate and 2 μl of 1 M sodium azide was spotted close to the test or solvent spot to immobilize worms once they have made their choice. The assay plates were incubated at 25°C, and the position of worms in the test zone, center zone, and control zone (Fig S1D) was noted at 2 h. The chemotaxis index was calculated by the formula: *Chemotaxis Index (CI) = (Worms in test zone − Worms in control zone)/Total number of worms*. Each data point in all the chemotaxis experiments represents one assay plate with 40–60 worms each.

For volatile-mediated chemotaxis: In a two-plate arrangement, 2 μl of test or solvent was spotted on two sides of the lower plate and approximately 40–60 young adult worms were washed and placed in the center of the upper plate and allowed to move to test area or control area on the basis of volatile cues (Fig S2A). Scoring was done after 2 h by counting the number of worms in the test area, and the chemotaxis index was calculated by the formula: *Chemotaxis Index (CI) = (Worms in test area − Worms in control area)/Total*.

### Bacterial odor choice assay

Bacterial odor choice was done in a tripartite plate with SK agar in two sectors. 50 μl of overnight grown bacterial cultures were spotted in one sector at 0 h and in the neighboring sector at 16 h and incubated at 37°C for another 8 h (Fig 1F). Young adult worms were washed twice with S-basal and once with assay buffer. 60–80 worms were placed in the third sector with buffered agar and incubated at 25°C. Worms attracted by the odor reached the barrier between the sectors and got immobilized there due to the presence of sodium azide. At 2 h, choice index was calculated by counting the number of worms in individual area facing the test side (young lawn) or control side (old lawn), and the choice index was calculated by the formula: *Choice Index (CI) = (Worms toward test side − Worms toward control side)/Total*.

### *C. elegans* survival assay

The survival assay was modified from a previous study (Singh & Aballay, 2006). 50 μl of overnight grown bacterial culture was spread uniformly on 60 mm SK plates and incubated at 37°C for 12 h. 50 young adult worms were placed in the center of the lawn, incubated at 25°C, and were scored for survival after 24 h for every 4 h. For "half lawn" survival experiments, 50 μl of the bacterial culture was spread in the center (20 mm diameter) on a 60-mm Petri dish. For odor pre-exposure, worms were exposed to 1-undecene (3 μl) for 2 h followed by 2 h of no odor, while naive worms were exposed to no odors for the entire duration of 2 h. Following this, the worms were exposed to full lawn of *P. aeruginosa* and scored for survival at 20°C. Time to death of 50% of the worms ($TD_{50}$) was calculated from three or more Kaplan–Meier survival curves. The curves were analyzed using the log-rank (Mantel–Cox) test. Each data point in all the survival experiment represents one assay with 90–100 worms each.

### Quantification of intestinal lumen width

Young adult N2 worms were exposed to *E. Coli* OP50 lawn and *P. aeruginosa* lawn in the same bacterial lawn size setup as done in the aversion response assay and incubated at 25°C for 12 h. The

worms were then randomly picked from inside or outside of the lawn; washed with M9 buffer containing 50 mM sodium azide, and mounted on 2% agar pads. The worms were visualized using Leica DMi8 inverted fluorescence microscope.

### SPME-GC-MS/MS analysis of volatiles produced by *P. aeruginosa*

An overnight culture of *P. aeruginosa* PA14 or mutant was used to seed with 50 µl spots on a 60-mm SK plate. The plates were then incubated for 22 h at 37°C. For the collection of volatiles, 2 *P. aeruginosa* PA14 or mutant spotted plates were sealed together using parafilm. The SPME fiber was inserted in between the two bacterial plates by puncturing a hole in parafilm between the plates with a needle (Fig S1A). A solid-phase microextraction fiber (SPME; divinylbenzene/carboxen/polydimethyl siloxane, 50/30 µm; Supelco, Sigma-Aldrich, Cat No. 57328-U) was used for the collection of volatiles. The fiber was exposed to the bacterial volatile for 1 h at room temperature. Immediately after collection, the SPME fiber was inserted into the GC injection port for desorption of the bacterial volatiles. The analysis was performed using 6890C gas chromatography (Agilent) interfaced with a 7000C mass selective detector (GC-MS). A capillary column, DB5-MS capillary column (30 m × 0.25 mm I.D. and film thickness 0.25 m, Agilent, Palo Alto, CA, USA), was used for separation with ultra-pure helium gas as the carrier gas at a constant flow rate of 3 ml/min. The injector was kept in split mode with a split ratio of 50:1. The column temperature program consisted of injection at 40°C, hold for 1 min, a temperature increase of 5°C/min to 170°C, followed by a temperature increase of 100°C/min to 270°C and hold for 2 min. The temperatures of the injector and MS source were maintained at 225°C and 265°C, respectively. Volatiles were identified by comparing the mass spectra obtained with the mass spectral library of the GCMS data system, NIST 11 (National Institute of Standards and Technology) mass spectral library.

### Calcium imaging

Calcium imaging in individual olfactory neurons expressing GCaMP (Appendix Table S1) was performed in a custom-designed microfluidic device (Chalasani *et al*, 2007). Transgenic worm expressing GCaMP family of genetically encoded calcium indicator in individual odor sensory neurons was trapped in the device, and their calcium response was assayed under different dilutions of 1-undecene. The dilutions of the chemicals were made using M9 buffer (5 g NaCl, 3 g $KH_2PO_4$, 6 g $Na_2HPO_4$, 1 ml 1 M $MgSO_4$ per liter of water). GCaMP imaging was performed using a Zeiss inverted microscope using a Photometrics EMCCD camera. The imaging was performed by capturing stacks of TIFF files for 180 s at 10 frames per s using Metamorph software. In 180 s imaging session for each worm, the chemical stimulus was provided in the sequence-10 s stimulus OFF, 120 s stimulus ON followed by 50 s stimulus OFF state. The images were analyzed using MATLAB scripts to plot the change in fluorescence to the baseline $F_o$ values. Bar diagrams were plotted as the average change in 10 s window after stimulus addition (time 11–20 s) and 10 s window after stimulus removal condition (time 131–140 s).

### Worm tracking and mean square displacement analysis

We designed a simple imaging setup for recording the motion of worms in the presence or absence of 1-undecene odor on agar plate. We took 15-min (12,286 frames) long recording of worms ($N = 5$) to track their behavior. Using MATLAB Image processing toolbox, we converted recorded frames into set of binary images, and then, using Trackmate, an ImageJ plugin, we tracked their motion. To analyze the obtained tracks, we used ensemble-averaged mean square displacement (MSD), $\langle \delta r^2(t) \rangle$ (Bewerunge *et al*, 2016) which helps in characterizing the dynamics of the worms. We then estimated the slope ($\mu$) of double logarithm MSD curve. MSD is defined by $\langle \delta r^2(t) \rangle \propto t^\mu$, where $\delta r$ is displacement of worms in time t and the slope of double logarithm MSD curve is defined by the slope $\mu$. For diffusion $\mu = 1$, for sub-diffusion $\mu < 1$ and for super-diffusion $\mu > 1$. In the analysis of video obtained from data (Fig 2E), we found that µ for control condition was 0.71 and µ under 1-undecene exposure was 0.92 (Fig S2B), which shows that the worms under 1-undecene odor exposure covered more area as compared to control condition.

### Gene expression analysis

Around 800–1,000 worms of L4 stage were exposed to the odor of 1-undecene provided by inverting 60 mm NGM plate with four spots of 3 µl 1-undecene used on the lid. Blank NGM plates were used as lids for control or naive worms. After 2 h of exposure at 25°C, worms were collected from plates by washing with M9 buffer and frozen in QIAzol lysis reagent at −80°C. Total RNA was extracted from the 1-undecene-exposed and naive worms using RNeasy plus universal mini kit (Qiagen, Cat. No. 73404), followed by DNase I (Thermo Scientific Cat. No. EN0525) treatment to remove genomic DNA. cDNA was synthesized using the iScript cDNA synthesis kit (Biorad, Cat. No. 170-8891) and used in real-time PCR for gene expression analysis using SYBR Green detection (Biorad Cat. No. 1725124) on quantStudio3 (Applied Biosystems) machine. All the Ct values were normalized to actin-1. The comparative ΔCt method was used to determine the fold change of each target gene. Primer sequences are available upon request.

For RNA isolation from the *P. aeruginosa* PA14, overnight culture was prepared from a fresh streaked plate. 500 µl of each culture was spot dried in SK plate. The plates were incubated at 37°C for 24 h for old lawn and 8 h for young lawn. The lawn from the plate was washed with PBS buffer, and 2 ml of 1 $OD_{600}$ culture was taken for RNA isolation for each sample. RNA isolation was done by hot acid phenol–chloroform method, modified from (Singh *et al*, 1995).

### Microscopy

Adult *irg-1p*::GFP transgenic worms containing *myo-3p*::mCherry co-marker in the pharynx were exposed to 1-undecene odor or control for 2 h as described in the previous section. For *P. aeruginosa exposure*, worms were allowed to feed on 12-h old complete lawn of PA14 at 25°C for 4 h. Adult *fmo-2p*::GFP transgenic worms were allowed to feed on 12-h old complete lawn of *E. coli* OP50, *E. faecalis* or *P. aeruginosa* for 8 h and on *E. coli* OP50 with 1-undecene exposure for 2 h. Adult *hsp-16.2p*::GFP worms were exposed to lawn of *E. coli* OP50 (naive), or lawn of *E. coli* OP50 with 1-undecene odor exposure for 2 h. For heat shock, worms placed on a lawn of *E. coli* OP50 were incubated at 37°C for 2 h followed by recovery for 4 h at 20°C. Adult *gst-4p*::GFP worms were exposed to lawn of *E. coli* OP50 (naive), or lawn of *E. coli* OP50 with

1-undecene odor exposure from lid for 2 h. For oxidative stress, worms were placed on *E. coli* OP50 lawn on an NGM plate with 20 mM paraquat and incubated for 5 h at 25°C.

In all conditions, worms were washed off with M9 buffer after treatment followed by suspending them in M9 buffer containing 50 mM sodium azide and mounting on 2% agar pads. The worms were visualized using an Olympus IX71 inverted fluorescence microscope.

## Statistical analysis

All statistical analysis was done using GraphPad Prism. Statistical analysis was done either by two-tailed unpaired *t*-test or one-way ANOVA, followed by Dunnett's multiple comparison test (mentioned in individual figure legends). The Kaplan–Meier method was used to calculate survival fractions and the log-rank test to compare survival curves. The significance annotations according to $P$ values —ns (not significant) $P > 0.05$; *$P \le 0.05$; **$P \le 0.01$; ***$P \le 0.001$; ****$P \le 0.0001$. All the experiments were repeated at least three times unless otherwise indicated.

## Data availability

This study includes no data deposited in external repositories.

**Expanded View** for this article is available online.

## Acknowledgements

We thank Emily Troemel for some *C. elegans* strains. Some *C. elegans* strains were provided by CGC which is funded by the NIH Office of Infrastructure Programs (P40 OD01440). We thank Anjali Gupta for creating AWA and ASH ablation lines, Divakar Badal for generating the mean square displacement curve. This work was supported by the Wellcome Trust/DBT India Alliance Intermediate Fellowship (Grant no. IA/I/13/1/500919) awarded to Varsha Singh. Some funding support was provided by DBT-IISC Partnership Program (BT/PR27952/INF/22/212/2018).

## Author contributions

RV, SHC, and VS supervised the project. DP and VS designed the study and performed the analysis. DP performed aversion, survival, choice, chemotaxis, gene expression analysis experiments; DP and BR performed GC-MS/MS experiments for volatile analysis under the supervision of RV; DP performed the GCaMP imaging under the supervision of SHC; DP and AMS performed Exploratory behavior experiment; DP, SHC and VS wrote the manuscript.

## Conflict of interest

The authors declare that they have no conflict of interest.

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
