## [Review Process File · The EMBO Journal]

1-Undecene from *P. aeruginosa* is an olfactory signal for flight or fight response in *C. elegans*

Deep Prakash, Akhil MS, Buddidhathi Radhika, Radhika Venkatesan, Sreekanth Chalasani, and Varsha Singh

DOI: [10.15252/embj.2020106938](https://doi.org/10.15252/embj.2020106938)

Corresponding author(s): Varsha Singh (varsha@iisc.ac.in)

Review Timeline:

Submission Date:	1st Oct 20
Editorial Decision:	11th Nov 20
Revision Received:	13th Feb 21
Editorial Decision:	25th Mar 21
Revision Received:	6th Apr 21
Accepted:	15th Apr 21

Editor: Karin Dumstrei

Transaction Report:

Dear Dr. Singh,

Thank you for submitting your manuscript to The EMBO Journal. Your study has now been seen by three referees and their comments are provided below.

As you can see from the comments, the referees find the analysis interesting. However significant revisions are also needed in order to consider publication here. Important controls are missing, some of the data needs to be better substantiated and further insight into how the induction of the host immune response pathway is linked to survival is needed. Should you be able to extend the findings along the lines suggested by the referees then I would be willing to consider a revised version.

I am happy to discuss the raised points further and maybe it would be most helpful to do so via phone or video.

When preparing your letter of response to the referees' comments, please bear in mind that this will form part of the Review Process File, and will therefore be available online to the community. For more details on our Transparent Editorial Process, please visit our website:

<https://www.embopress.org/page/journal/14602075/authorguide#transparentprocess>

Thank you for the opportunity to consider your work for publication. I look forward to discussing your revisions further with you

Yours sincerely,

Karin Dumstrei, PhD
Senior Editor
The EMBO Journal

Further information is available in our Guide For Authors:

The revision must be submitted online within 90 days; please click on the link below to submit the revision online before 9th Feb 2021.

Referee #1:

In this manuscript, the authors report that a *Pseudomonas* specific volatile odor elicits behavioral avoidance and induces immune-specific gene expression, increasing survival in response to pathogenic bacteria. There is evidence that host immune responses may be driven by chemosensory cues in multiple systems, although the molecular mechanisms of these responses are largely unknown. Recently, it was demonstrated that another *C. elegans* olfactory neuron - AWC - plays a critical role in non-autonomous regulation of p38 MAPK dependent immune responses via unidentified odors (Foster et al., 2020). Thus, the identification of a parallel olfactory input, including the identity of both the odors and molecular pathways involved, would be a really novel finding of broad interest.

The authors show that 1-undecene elicits behavioral avoidance and induces immune response genes and that undecene production by *Pseudomonas* is necessary for these responses. This is a really interesting finding, however, at this point there are many cases in which missing controls and/or an insufficiently fleshed out pathway severely limits the interpretation of these data. Additionally, it is unclear how the innate immune responses shown here are coordinated with the observed acute behavioral responses to alter survival.

Specific comments:

1. A major concern is the lack of both genetic and chemical controls for most of the experiments. For all of the *C. elegans* mutants, only a single allele was analyzed and no rescue experiments were performed. This is important to not only control for genetic background, but also to confirm the specificity of the proposed AWB neuron function. For example, *odr-3* is expressed in AWB and AWC, as noted by the authors, but also in ASH, where it is necessary for odor responses (see Yoshida et al., 2012). How specific are the observed responses for 1-undecene? Can the authors use an analogous volatile chemical that does not activate AWB in the imaging and gene induction experiments? □

2. The effects on foraging behavior are interesting but are a bit confusing. Do these locomotory changes lead to increased lawn-leaving behavior? The increased survival on a partial lawn of PA14 suggests they should.

3. How does the 1-undecene dependent induction of innate immune response genes relate to the survival of the worms on PA14? The authors propose that 1-undecene acts via the *zip-2* pathway to induce *irg-1/2/3* expression. *zip-2* mutants have decreased survival on PA14 lawns, but the authors only observed a survival effect on partial lawns. Does this mean the immune response induction does not alter survival and is secondary to the behavioral response? Is survival reduced on *undA* mutant bacteria and does *undA* play a role in virulence? A major question in the field is how chemosensory and innate immune responses are coordinated to alter survival and in my opinion, this could be further fleshed out. □

4. Gene expression experiments: for the RT-PCR experiments, it is unclear how many replicates were performed or how the experiments were designed. For example, it appears that Figure 1A and 1C contain the same data for *irg-1/2/3*, if this is the case it should be explicitly noted. If Figure 1A was essentially exploratory in nature, then in my opinion, the control data should have been replicated in order to test a different hypothesis in 1C. An alternative might be to use reporters, as in 4B, but including the appropriate controls to demonstrate an AWB specific effect. Similar controls should be performed to compare the wild-type vs. *undA* mutants.

Minor suggestions:

1. The chemotaxis effect in the bacterial *undA* mutant are striking, and it seems odd that these data are not included in the main figures.
2. There are several grammatical errors in the manuscript which could use a little more proofreading.
3. I could not find a methods entry for the acute avoidance experiments and what, if any, controls were done for this experiment?
4. Figure 3C-F all seem to show essentially the same data, perhaps this could be made more concise?
5. Line 292 makes reference to "...aversion response in at least 80% of the population" - it's unclear to what this is referencing. □

Referee #2:

In this study "1-Undecene from *P. aeruginosa* is an olfactory signal for flight or fight response in *C. elegans*", the authors identified 1-undecene as a volatile product of a type of infectious

Pseudomonas aeruginosa. They showed that 1-undecene was repulsive to *C. elegans* worms and *Pseudomonas aeruginosa* unable to produce 1-undecene could not repel *C. elegans*. 1-undecene stimulates GCaMP3 signals in AWB neurons, and repulsion of 1-undecene to *C. elegans* is lost in a *lim-4* mutant. Exposure to 1-undecene induces expression of several *zip-2*-dependent immune response genes. Overall, the authors aim to address an interesting question. The experiments are well designed, and the results are likely to contribute to a better understanding of host-pathogen interactions. Addressing or clarifying the following several questions will help to improve the manuscript.

1. The authors used an *odr-3* mutant to examine the function of odor sensation in their analysis. In addition to odor sensory neurons, *odr-3* is also expressed in other chemosensory neurons including nociceptive neurons ASH, phasmid neurons PHB. The results based on testing *odr-3* are not fully conclusive about the involvement of odor sensation. It is helpful that authors used a *lim-4* mutant to strengthen the conclusion. However, *lim-4* is found in many neurons including motor neurons. The movement defects of the *lim-4* mutant may interfere with chemotaxis assays. An AWB-specific ablation line or a similar reagent is needed to support the function of AWB in this study.
2. Because one single mutant was used for *odr-3* and for *lim-4*, the conclusion based on the current results needs to be confirmed with rescue. Or, the authors should at least test a second mutation for each of these genes.
3. Figure 2b shows that when 1-undecene is tested alone, it repels worms. However, it is not clear whether 1-undecene repels worms when it is a part of the volatile products of a bacterial lawn. The authors should test whether 1-undecene added to a lawn of *E. coli* repels worms and quantify the lawn leaving behavior similarly as in figure 1a and 1b. In figure 2d the authors showed trajectories of worms. However, it is difficult to see on these panels where 1-undecene is and where the lawn borders are. In addition, the results need to be quantified.
4. "We observed increased speed and enhanced roaming by worms in an arena exposed to 1-undecene odor compared to the arena without the repellent (Fig. 2D and S2B). Worms also executed omega bend in response to 1-undecene in 16 out of 17 worms (Figure 2C, Supplementary Movie S3) consistent with response to an aversive signal." The speed and the omega bend need to be quantified and compared with controls.
5. T-test is perhaps not the best statistical test for the data in fig 1e, 3b and 3d.
6. Is 1-undecene a microbe-associated molecular pattern or a pathogen-associated molecular pattern? The authors propose the later, because exposure to 1-undecene induces expression of *zip-2*-dependent immune genes. If this is the case, mutants of PA14 with weaker virulence would not produce 1-undecene. Did author test this possibility? Clarification of this question will better characterize the property and function of 1-undecene.
7. "Moreover, we also could not detect 1-undecene in *E. coli* OP50, *Salmonella typhimurium*, and *Enterococcus faecalis* headspace (data not shown)". It will be very helpful to include these data.
8. It is intriguing that exposure to 1-undecene specifically upregulates *zip-2*-dependent immune genes *irg-1,2,3*. Does the induction depend on *zip-2*?
9. The logic for line 47-65 should be better clarified. The authors summarized several recent studies that addressed fight (immune response) and flight (avoidance and learning) responses to pathogens, odorant sensation of pathogens, virulence-independent sensation, and anatomical changes caused by pathogen exposure. It is useful for the authors to better organize these information and clarify how some or all of these studies relate to their question "However, the nature of *P. aeruginosa* volatile molecules that facilitate pattern recognition and threat perception in *C. elegans* remain unclear."
10. "However, worms have a large sensory repertoire of 302 neurons..." This sentence is not accurate.

Referee #3:

Through an analytical chemistry approach, Prakash et al. identify 1-undecene as a volatile produced by *P. aeruginosa* that can alter behavior and gene expression in the nematode *C. elegans*. They present data suggesting that it acts via the neuron AWB. The responses of *C. elegans* to pathogens remains an area of active research and these results will potentially interest those working in the field.

Most of the data appears robust and generally the study is well presented. There are some recent studies that should have been cited, including Ringstad's on O₂/CO₂ sensing, and most pertinently, "Identification of Odor Blend Used by *Caenorhabditis elegans* for Pathogen Recognition", Worthy et al. (2018) in the specialist journal, *Chemical Senses*.

The most significant claim that Prakash et al. make, "1-undecene serves as a molecular pattern and induces upregulation of a subset of immune response genes specific to *P. aeruginosa* in worms", is currently not sufficiently supported by the data.

To demonstrate that 1-undecene has a specific effect on host gene expression, indicative of a pathogen-specific immune response, they assayed the expression of genes that are induced by *P. aeruginosa*, including *irg-1* and *irg-2*, and genes that are not induced by *P. aeruginosa*. Work from the Troemel lab and others, has shown that *irg-1* and *irg-2* are far from being pathogen-specific markers. They have been suggested to be part of an "Intracellular Pathogen Response" genes (Reddy et al 2017), regulated by *pals-22* and *pals-25* (Reddy et al. 2019), and induced by a range of different pathogens (e.g Orsay virus for *irg-2*), as a consequence of cellular dysfunction caused by infection. This is compatible with the fact *irg 1* and *irg 2* expression is strongly induced by the translational elongation inhibitor cycloheximide (Dunbar 2012). 1-undecene is far from a biologically inert molecule; it is known to have antimicrobial activity for example (10.3389/fmicb.2015.01082 and references therein). Its effects on host gene expression could reflect disruption of cellular homeostasis.

If the authors want to support their suggestion of a pathogen-specific response, they need to demonstrate that the 1-undecene-associated increases in gene expression are independent of *pals* regulation. They should also examine the expression of different markers of cellular stress. Further, they should assay the expression of candidate genes that they recently identified as being regulated by *P. aeruginosa*, *E. faecalis* and *C. neoformans*, such as *dct-17* and *lys-3*.

Other minor concerns that should be addressed:

Figure 1F: N2 appear to have 2 distinct populations, one responsive and one un-responsive. There is a similar separation in Figure S2D. What is the explanation?

Welch's t-test is insensitive to equality of the variances, but does assume normal distribution. It should not be used for non-normal data (like Figure 1F).

If the control data for Figure 4C is identical to the data in Figure 4A, this needs to be indicated.

qRT-PCR is presented, for example, as "Relative mRNA expression N2 (1-Undecene / Naive)", but there are negative values, so this cannot be right. What is actually being represented?

The authors make a distinction between "younger" and "older" lawns. If 1-undecene is not produced by *P. aeruginosa* in liquid culture and/or at 37C, the differences observed would simply be a question of the time that the bacterial cultures have been in a situation compatible with 1-undecene synthesis. This could readily be checked by assaying undA expression under the different culture conditions.

Referee #1:

In this manuscript, the authors report that a *Pseudomonas* specific volatile odor elicits behavioral avoidance and induces immune-specific gene expression, increasing survival in response to pathogenic bacteria. There is evidence that host immune responses may be driven by chemosensory cues in multiple systems, although the molecular mechanisms of these responses are largely unknown. Recently, it was demonstrated that another *C. elegans* olfactory neuron - AWC - plays a critical role in non-autonomous regulation of p38 MAPK dependent immune responses via unidentified odors (Foster et al., 2020). Thus, the identification of a parallel olfactory input, including the identity of both the odors and molecular pathways involved, would be a really novel finding of broad interest.

The authors show that 1-undecene elicits behavioral avoidance and induces immune response genes and that undecene production by *Pseudomonas* is necessary for these responses. This is a really interesting finding, however, at this point there are many cases in which missing controls and/or an insufficiently fleshed out pathway severely limits the interpretation of these data. Additionally, it is unclear how the innate immune responses shown here are coordinated with the observed acute behavioral responses to alter survival.

Specific comments:

1. A major concern is the lack of both genetic and chemical controls for most of the experiments. For all of the *C. elegans* mutants, only a single allele was analyzed and no rescue experiments were performed. This is important to not only control for genetic background, but also to confirm the specificity of the proposed AWB neuron function. For example, *odr-3* is expressed in AWB and AWC, as noted by the authors, but also in ASH, where it is necessary for odor responses (see Yoshida et al., 2012). How specific are the observed responses for 1-undecene? Can the authors use an analogous volatile chemical that does not activate AWB in the imaging and gene induction experiments?

RESPONSE: We agree with the reviewer that ODR-3 is also expressed non-olfactory neurons such as nociceptive neuron ASH which can respond to odors (Yoshida et al., 2012). In the revised manuscript, we have used additional allele for *odr-3(n2046)*, in addition to *odr-3(n2150)* as well as *lim-4(yz12)* in addition to *lim-4(ky403)*. These experiments are described in response to the next comment. We have also used ablation for ASH neurons to show that this neuron is

not involved in response to 1-undecene (Figure 3B). Additionally, we also analysed GCaMP response of ASH neurons to 1-undecene and found no response (Rebuttal Figure 1). To control for volatile chemical stimuli, we have studied the transcriptional response of *C. elegans* upon 2 hours exposure to diacetyl (2,3 butanedione), an odor sensed by AWA olfactory neurons. We found no significant upregulation of *irg-1*, *irg-2* or *irg-3* (shown in rebuttal Figure 2 A and 2B) by qRT-PCR or using *irg-1::GFP* reporter..

Rebuttal Figure 1

Rebuttal Figure 1: Heat map of calcium response in ASH::GCaMP3. Each row represents an individual worm recorded for 180 s under 1:100 dilution of 1-undecene, in 11 s-130 s window.

Rebuttal Figure 2

Rebuttal Figure 2: (A) Real time PCR analysis of *irg-1*, *irg-2* and *irg-3* genes in naive and 1-undecene odor exposed N2 worms. n = 3. Error bars indicate SEM. (B) *irg-1p::GFP* induction in naive worms and worms exposed to diacetyl odor (3 μ l, 4 spots, 2 h).

2. The effects on foraging behavior are interesting but are a bit confusing. Do these locomotory changes lead to increased lawn-leaving behavior? The increased survival on a partial lawn of PA14 suggests they should.

RESPONSE: We believe that locomotory behavior is likely a search for area with lower concentration of 1-undecene. And yes, we agree with the reviewer that avoidance is a survival strategy for worms as our experiments with partial and full lawn and with *odr-3* mutants (Figure 1 in the revised manuscript) would indicate.

3. How does the 1-undecene dependent induction of innate immune response genes relate to the survival of the worms on PA14? The authors propose that 1-undecene acts via the *zip-2* pathway to induce *irg-1/2/3* expression. *zip-2* mutants have decreased survival on PA14 lawns, but the authors only observed a survival effect on partial lawns. Does this mean the immune response induction does not alter survival and is secondary to the behavioral response? Is survival reduced on *undA* mutant bacteria and does *undA* play a role in virulence? A major question in the field is how chemosensory and innate immune responses are coordinated to alter survival and in my opinion, this could be further fleshed out. [SEP]

RESPONSE: Thanks for your suggestions.

We have pre-exposed worms to 1-undecene before survival assays on *P. aeruginosa*. As shown in Figure 4 (new panel added in Figure 4F and 4G of the revised manuscript), pre-exposure to 1-undecene enhances the survival of worms from *P. aeruginosa* compared to naive worms. The experiment is shown as rebuttal Figure 3 here.

If 1-undecene is indeed a molecular pattern necessary for induction of protective immune response in worms, we expected to see reduced survival of worms on *undA* compared to on wild type *P. aeruginosa* PA14. As shown in rebuttal Figure 4 *C. elegans* survival is indeed reduced on *undA* mutant. Since we observed induction of *irg-1*, *irg-2* and *irg-3* within two hours of exposure, we would like to classify it as an immediate-early response.

Rebuttal Figure 3

Rebuttal Figure 3: Preexposure of worms to 1-undecene enhances the survival of worms upon subsequent infection with *P. aeruginosa*.

- (A) Kaplan Meier survival curve on *P. aeruginosa* for N2 (naive) worms and N2 worms pre-exposed to 1-undecene odor. Survival assay was performed at 20°C.
- (B) Time required for 50% of worms to die (TD₅₀) on *P. aeruginosa*. Each data point indicates replicates with ~100 worms. n = 3 assays. ** P ≤ 0.01 as determined by two-tailed unpaired t-test. Error bars indicate SEM.

Rebuttal Figure 4

Rebuttal Figure 4:

- (A) Kaplan Meier survival curve of N2 worms on *P. aeruginosa* wild type (PA14) and *undA* mutant. Survival assay was performed at 20°C.
- (B) Time required for 50% of worms to die (TD₅₀) on *P. aeruginosa* wild type (PA14) and *undA* mutant. Each data point indicates replicates with ~100 worms. n = 3 assays. * P ≤ 0.05 as determined by two-tailed unpaired t-test. Error bars indicate SEM.

4. Gene expression experiments: for the RT-PCR experiments, it is unclear how many replicates were performed or how the experiments were designed. For example, it appears that Figure 1A and 1C contain the same data for *irg-1/2/3*, if this is the case it should be explicitly noted. If Figure 1A was essentially exploratory in nature, then in my opinion, the control data should have been replicated in order to test a different hypothesis in 1C. An alternative might be to use reporters, as in 4B, but including the appropriate controls to demonstrate an AWB specific effect. Similar controls should be performed to compare the wild-type vs. *undA* mutants.

RESPONSE: We think the reviewer is referring to qRT-PCR analyses in Figure 4. We have taken care to describe the number of biological replicates for each experiment in the methods section of the revised manuscript. All the control data (N2 worms exposed to 1-undecene odor) has been replicated several times for the revision and included in revised Figure 4. We have ensured that control data (WT worms exposed to 1-undecene) is from different experiments across panels in Figure 4. We have also analysed the transcriptional response of 1-undecene exposure in two alleles of *odr-3* and 2 alleles of *lim-4* in the revised Figure 4.

Minor suggestions:

1. The chemotaxis effect in the bacterial *undA* mutant are striking, and it seems odd that these data are not included in the main figures.

RESPONSE: We have included the choice assay for *undA* mutant (over wild type PA14) in main Figure 2 in the revised manuscript.

2. There are several grammatical errors in the manuscript which could use a little more proofreading.

RESPONSE: We have proofread the entire manuscript.

3. I could not find a methods entry for the acute avoidance experiments and what, if any, controls were done for this experiment?

RESPONSE: For chemotaxis assays, standard protocol was used with some modifications (Bargmann et al., 1993). The details are included in the Methods section and Supplementary Figure S1D of the revised manuscript. As a control, we performed chemotaxis assays against a well-studied repellent 2-nonanone and attractant diacetyl. The data for controls is shown here (rebuttal Figure 5).

Rebuttal Figure 5

Rebuttal Figure 5:

Chemotactic response of N2 worms towards diacetyl and 2-nonanone. Each data point indicates an individual assay plate with ~40 worms. $n \geq 3$ assays. Error bars indicate SEM.

4. Figure 3C-F all seem to show essentially the same data, perhaps this could be made more concise?

RESPONSE: Thanks for your comments. Since our study is the first report on 1-undecene as a repellent and the first report of stimulation for AWB neurons, all the panels provide useful information for the *C. elegans* community. Therefore, we would like to retain all the panels.

5. Line 292 makes reference to "...aversion response in at least 80% of the population" - it's unclear to what this is referencing. [L]
[SEP]

RESPONSE: We Have rephrased the statement for better understanding.

"A large fraction of worms in a population showed aversion response suggesting that 1-undecene is a physiologically relevant stimulus for worms."

Referee #2:

In this study "1-Undecene from *P. aeruginosa* is an olfactory signal for flight or fight response in *C. elegans*", the authors identified 1-undecene as a volatile product of a type of infectious *Pseudomonas aeruginosa*. They showed that 1-undecene was repulsive to *C. elegans* worms and *Pseudomonas aeruginosa* unable to produce 1-undecene could not repel *C. elegans*. 1-undecene stimulates GCaMP3 signals in AWB neurons, and repulsion of 1-undecene to *C. elegans* is lost in a *lim-4* mutant. Exposure to 1-undecene induces expression of several *zip-2*-dependent immune response genes. Overall, the authors aim to address an interesting question. The experiments are well designed, and the results are likely to contribute to a better understanding of host-pathogen interactions. Addressing or clarifying the following several questions will help to improve the manuscript.

1. The authors used an *odr-3* mutant to examine the function of odor sensation in their analysis. In addition to odor sensory neurons, *odr-3* is also expressed in other chemosensory neurons including nociceptive neurons ASH, phasmid neurons PHB. The results based on testing *odr-3* are not fully conclusive about the involvement of odor sensation. It is helpful that authors used a *lim-4* mutant to strengthen the conclusion. However, *lim-4* is found in many neurons including motor neurons. The movement defects of the *lim-4* mutant may interfere with chemotaxis assays. An AWB-specific ablation line or a similar reagent is needed to support the function of AWB in this study.

RESPONSE: We agree with the reviewer that ODR-3 is expressed in non-olfactory neurons such as nociceptive neuron ASH which can respond to odors. We have used ablation for ASH neurons to show that this neuron is not involved in response to 1-undecene (Figure 3B in the revised manuscript). We also analyzed GCaMP response of ASH neurons to 1-undecene and found no response (Rebuttal Figure 1).

In the revised manuscript, we have used an additional allele *odr-3(n2046)*, in addition to *odr-3(n2150)*. Our efforts to create AWB ablation strain have failed so far and we have not found viable ablation lines. Therefore, we have utilized an additional allele of *lim-4(yz12)* in addition to *lim-4(ky403)* and were able to phenocopy all the phenotypes.

2. Because one single mutant was used for *odr-3* and for *lim-4*, the conclusion based on the current results needs to be confirmed with rescue. Or, the authors should at least test a second mutation for each of these genes.

RESPONSE: Thank you. We have used additional allele *odr-3(n2046)*, in addition to *odr-3(n2150)* as well as *lim-4(yz12)* in addition to *lim-4(ky403)*. We have included data for both the alleles of *odr-3* in Figure 1 in the revised manuscript. This includes lawn leaving data (Figure 1A), survival assays (Figure 1 C-E). Transcriptional response for both alleles of *odr-3* is included in Figure 4D in the revised manuscript.

We have included two alleles of *lim-4* in 1-undecene chemotaxis assays (Figure 3B) and on the transcriptional response to 1-undecene exposure (Figure 4E). Both alleles fail to show induction of *irg-1*, *irg-2* and *irg-3* in response to 1-undecene.

3. Figure 2b shows that when 1-undecene is added alone, it repels worms. However, it is not clear whether 1-undecene repels worms when it is a part of the volatile products of a bacterial lawn. The authors should test whether 1-undecene added to a lawn of *E. coli* repels worms and

quantify the lawn leaving behavior similarly as in figure 1a and 1b. In figure 2d the authors showed trajectories of worms. However, it is difficult to see on these panels where 1-undecene is and where the lawn borders are. In addition, the results need to be quantified.

RESPONSE: Thanks for this suggestion. When 1-undecene was adsorbed in OP50 lawn, we could observe multiple reversals and omega turns on the lawn, as expected but we did not observe complete lawn leaving response.

We designed a different experiment to address if bacteria prefer OP50 lawn over OP50 lawn with 1-undecene. As described in the schematic in Rebuttal Figure 6, two OP50 lawns, A and B were separated by 5 mm. In test plates, 1-undecene was absorbed on lawn B or left as is in control plates. Worms were placed on lawn A and we examined if worms cross over to lawn B in 2 hours. In control plates, we found that worms moved to lawn B in all the control plates but only a small fraction of worms moved to lawn B in test plates. Based on this experiment, we inferred that *C. elegans* prefers OP50 food without 1-undecene.

Rebuttal Figure 6

Rebuttal Figure 6:

- (A) Schematic of the experimental setup used to record the repulsion behavior of worms from 1-undecene. Lawn A and B are *E. coli* OP50. Lawn B in the test condition is spiked with 0.5 μ l of 1-undecene (test lawn). Worms were introduced in lawn A at the start of the experiment and the distribution of worms on lawn B was recorded after an interval of 120 m in both control and test condition.
- (B) Percent distribution of worms in the test lawn (lawn B) in both control and test condition after an interval of 120 minutes. Each data point represents one assay plate with \sim 20 worms. $n \geq 3$ assays. *** $P \leq 0.001$ as determined by two-tailed unpaired t-test. Error bars indicate SEM.

4. "We observed increased speed and enhanced roaming by worms in an arena exposed to 1-undecene odor compared to the arena without the repellent (Fig. 2D and S2B). Worms also executed omega bend in response to 1-undecene in 16 out of 17 worms (Figure 2C, Supplementary Movie S3) consistent with response to an aversive signal." The speed and the omega bend need to be quantified and compared with controls.

RESPONSE: We have quantified omega turns in *C. elegans* exposed to 1-undecene or no odor for 5 minutes. All the worms exposed to 1-undecene odor executed omega turns (rebuttal figure 7A).

We have also quantified the speed of the worms under 1-undecene odor exposure and control condition (rebuttal figure 7B). The coordinate information of worms from 2604 tracking dataset was imported into MATLAB 2019B. The imported data was then used to calculate the distance

by using Euclidean distance formula. Distance obtained was converted into speed by dividing with time interval between two consecutive frames. The probability density histogram was generated (bin size of 0.05) for worms' track in the control and 1-undecene exposed conditions. Further, the significance was determined using Kolmogorov–Smirnov test. This data suggests that the average speed of worms exposed to 1-undecene was significantly higher compared to control.

Rebuttal Figure 7

Rebuttal Figure 7:

- (A) Percent of worms performing omega turn as observed under a window of 5 minutes. Each data point represents an average of 5 worms tested. n = 3. *** P ≤ 0.001 as determined by two-tailed unpaired t-test. Error bars indicate SEM.
- (B) Histogram of probability density distribution of the velocity threshold of worms exposed to 1-undecene odor or control condition. Significance was calculated using Kolmogorov–Smirnov test (P = 0.0144).

5. T-test is perhaps not the best statistical test for the data in fig 1e, 3b and 3d.

RESPONSE: Figure 1E has been analyzed with two-tailed, unpaired t-test (Ha et al., 2010; Styer et al., 2008).

Figure 3B and 3D in the revised manuscript have been analyzed using one-way ANOVA followed by Dunnett's multiple comparison test (Jang et al., 2019; Meisel et al., 2014; Singh and Aballay, 2019; Worthy et al., 2018).

6. Is 1-undecene a microbe-associated molecular pattern or a pathogen-associated molecular pattern? The authors propose the later, because exposure to 1-undecene induces expression of zip-2-dependent immune genes. If this is the case, mutants of PA14 with weaker virulence would not produce 1-undecene. Did author test this possibility? Clarification of this question will better characterize the property and function of 1-undecene.

RESPONSE: We hypothesize that *P. aeruginosa* strains with apparent increased virulence may not produce (or produce less of) 1-undecene resulting in dampened host immune response. We see reduced survival of *C. elegans* on *undA* mutant (rebuttal Figure 4) suggesting that 1-undecene is necessary for induction of immune response, consistent with our hypothesis. Additionally, we examined *undA* transcript level in a quorum-sensing defective and hypo virulent mutant *rhIR*. As shown in rebuttal Figure 8, we found no significant alteration in expression suggesting that UndA is not under the control of quorum. This suggests that there

could be several mechanisms for reduced virulence in *P. aeruginosa*, 1-undecene is just one of them and it is not altered in *rhlR* mutant.

Rebuttal Figure 8

Rebuttal Figure 8:

Real time PCR analysis of *undA* transcript in lawn of *rhlR* mutant (24 h) over lawn of PA14 (24 h). n = 3. Error bars indicate SEM.

7. "Moreover, we also could not detect 1-undecene in *E. coli* OP50, *Salmonella typhimurium*, and *Enterococcus faecalis* headspace (data not shown)". It will be very helpful to include these data.

RESPONSE: Thank you. We show data for *E. coli* OP50 and *Enterococcus faecalis* OG1RF (Rebuttal Figure 9). As shown, the 1-undecene peak at 14 minutes is not seen in either of these two bacteria. We also closely surveyed the literature for several bacteria (*Staphylococcus aureus*, *Salmonella Typhimurium*, *Serratia marcescens*) and found that none of these produced 1-undecene (Siripatrawan, 2008; Worthy et al., 2018; Zhu et al., 2010).

Rebuttal Figure 9

Rebuttal Figure 9:

(A) GC-MS/MS profile of volatiles produced by 24 h old lawn of *E. coli* OP50.

(B) GC-MS/MS profile of volatiles produced by 24 h old lawn of *E. faecalis* OG1RF.

8. It is intriguing that exposure to 1-undecene specifically upregulates zip-2-dependent immune genes *irg-1,2,3*. Does the induction depend on zip-2?

RESPONSE: Yes, the transcription of *irg-1* and *irg-2* is dependent in ZIP-2 transcription factor. We found that 1-undecene odor exposure for 2 hours failed to induce *irg-1* and *irg-2* in *zip-2(tm4248)* mutant. This is included in the revised manuscript as Figure 4B.

9. The logic for line 47-65 should be better clarified. The authors summarized several recent studies that addressed fight (immune response) and flight (avoidance and learning) responses to pathogens, odorant sensation of pathogens, virulence-independent sensation, and anatomical changes caused by pathogen exposure. It is useful for the authors to better organize these information and clarify how some or all of these studies relate to their question "However, the nature of *P. aeruginosa* volatile molecules that facilitate pattern recognition and threat perception in *C. elegans* remain unclear."

RESPONSE: We have reorganized this paragraph in the revised manuscript, to lead to the question of whether volatile molecules serve as microbe-associated molecular patterns.

10. "However, worms have a large sensory repertoire of 302 neurons..." This sentence is not accurate.

RESPONSE: We have rephrased to:

"However, worms have a large sensory repertoire of ~1300 G protein-coupled receptors."

Referee #3:

Through an analytical chemistry approach, Prakash et al. identify 1-undecene as a volatile produced by *P. aeruginosa* that can alter behavior and gene expression in the nematode *C. elegans*. They present data suggesting that it acts via the neuron AWB. The responses of *C. elegans* to pathogens remains an area of active research and these results will potentially interest those working in the field.

Most of the data appears robust and generally the study is well presented. There are some recent studies that should have been cited, including Ringstad's on O₂/CO₂ sensing, and most pertinently, "Identification of Odor Blend Used by *Caenorhabditis elegans* for Pathogen Recognition", Worthy et al. (2018) in the specialist journal, *Chemical Senses*.

RESPONSE: Thank you. Several additional references have been included in the revised manuscript including those mentioned by the reviewer. We have been unable to accommodate Ringstad et al, 2013 in our study as it is focused on CO₂ sensing neuron and not linked to pathogen recognition or secondary metabolite perception, to the best of our knowledge.

The most significant claim that Prakash et al. make, "1-undecene serves as a molecular pattern and induces upregulation of a subset of immune response genes specific to *P. aeruginosa* in worms", is currently not sufficiently supported by the data.

RESPONSE: In the revised manuscript, we have pre-exposed worms to 1-undecene volatile followed by analysis of survival on live *P. aeruginosa* lawn. We find that pre-exposure enhances resistance of worms proving evidence that immune response induction is linked to protection (new panel added in Figure 4, 4F and 4G in the revised manuscript). The experiment is shown as rebuttal Figure 3 here. *irg-1*, *irg-2* and *irg-3* are specific to *P. aeruginosa* as shown previously by Troemel lab (Estes et al., 2010).

We have also analysed 8 immune effectors specific to Gram-positive bacteria and pathogenic yeast and find that they are not upregulated by exposure to 1-undecene odor (Fig. S4D). 1-undecene also does activate heat shock response, oxidative stress response (Fig. S4 G-H in the revised manuscript), or Intracellular pathogen response (rebuttal Figure 10). Collectively these experiments provide evidence that 1-undecene odor induces specific protection against *P. aeruginosa* and does not induce other responses.

To demonstrate that 1-undecene has a specific effect on host gene expression, indicative of a pathogen-specific immune response, they assayed the expression of genes that are induced by *P. aeruginosa*, including *irg-1* and *irg-2*, and genes that are not induced by *P. aeruginosa*. Work from the Troemel lab and others, has shown that *irg-1* and *irg-2* are far from being pathogen-specific markers. They have been suggested to be part of an "Intracellular Pathogen Response" genes (Reddy et al 2017), regulated by *pals-22* and *pals-25* (Reddy et al. 2019), and induced by a range of different pathogens (e.g Orsay virus for *irg-2*), as a consequence of cellular dysfunction caused by infection. This is compatible with the fact *irg 1* and *irg 2* expression is strongly induced by the translational elongation inhibitor cycloheximide (Dunbar 2012). 1-undecene is far from a biologically inert molecule; it is known to have antimicrobial activity for example (10.3389/fmicb.2015.01082 and references therein). Its effects on host gene expression could reflect disruption of cellular homeostasis.

RESPONSE: Thank you. Troemel lab has shown that *irg-1*, *irg-2* and *irg-3* are specific to *P. aeruginosa* infection and translational inhibition by the pathogen (Dunbar et al., 2012; Estes et

al., 2010). We have carefully examined the literature on IPR in *C. elegans*. Intracellular pathogen response is induced in worms invaded by natural viruses and Microsporidia (Reddy et al., 2017; Reddy et al., 2019). We found no evidence in the literature that *irg-1*, *irg-2*, *irg-3* or *zip-2* transcription are components of IPR in *C. elegans*. They are indeed shown to be induced only in response to *P. aeruginosa* (Estes et al., 2010; Troemel et al., 2006).

Additionally, we examined if 1-undecene exposure can induce IPR response by looking at an IPR reporter *pals-5p::GFP*, obtained from Troemel lab at UCSD. As shown in rebuttal Figure 10, 1-undecene exposure did not induce *pals-5p::GFP* expression. We further confirmed that *P. aeruginosa* live bacteria also do not induce IPR reporter *pals-5* (rebuttal Figure 10). Since *pals-5* was not regulated by either *P. aeruginosa* or 1-undecene, we did not examine upstream regulators of *pals-5* such as *pals-22* or *pals-25*. To confirm that the strain was behaving normally, we used heat shock at 30°C and found that *pals-5p::GFP* expression was induced in our strain as reported by Troemel lab (Reddy et al., 2017).

Rebuttal Figure 10

Rebuttal Figure 10:

pals-5p::GFP induction in worms exposed to *E. coli* OP50 (blank), *P. aeruginosa*, *E. coli* OP50 under 1-undecene odor exposure and heat shock at 30°C for 24 h.

Thanks for suggestion to carefully look at the effects of the bacterial volatiles, 1-undecene, DMDS etc, on plants. It has been shown in the literature that 1-undecene in combination with another volatile promotes plant growth, although molecular mechanisms remain to be deciphered. It is especially relevant because *P. aeruginosa* and other *Pseudomonas* species are associated with plants in nature. However, in fungus, *Phytophthora infestans*, 1-undecene has a negative impact. In the discussion section of the revised manuscript, we have discussed the effect of 1-undecene on plants and fungi. The evidence from published literature (Hunziker et al., 2015; Lo Cantore et al., 2015) and our study point to 1-undecene as a molecular pattern which mediates interkingdom interactions. These references are included in the Discussion section of the revised manuscript.

If the authors want to support their suggestion of a pathogen-specific response, they need to demonstrate that the 1-undecene-associated increases in gene expression are independent of

pals regulation. They should also examine the expression of different markers of cellular stress. Further, they should assay the expression of candidate genes that they recently identified as being regulated by *P. aeruginosa*, *E. faecalis* and *C. neoformans*, such as *dct-17* and *lys-3*.

RESPONSE: Please see the response to the previous comment. We found that IPR reporter *pals-5* was not upregulated by 1-undecene odor exposure (Rebuttal Figure 10). We also studied the activation of cytosolic stress response and oxidative stress response machinery upon exposure of worms to 1-undecene. As shown in rebuttal Figure 11, *hsp-16.2p::GFP* was induced by heat shock and recovery but not by 1-undecene odor exposure (Fig. S4G). Glutathione-s-transferase *gst-4p::GFP* was induced by exposure of worms to 20 mM paraquat but not by 1-undecene exposure (Fig. S4H in the revised manuscript and rebuttal Figure 12). Based on these experiments, we inferred that 1-undecene volatile does not cause disruption of cellular homeostasis.

We also examined the expression of additional genes upregulated during response of worms to *E. faecalis* and *C. neoformans*. We analysed *fmo-2*, *acs-2*, *lipl-1*, *lipl-3*, *cpr-4*, *cpr-5*, *asp-14*, *lys-3* and found that their transcripts were not significantly upregulated by 1-undecene odor exposure (Supplementary Figure S4D in the revised manuscript).

Rebuttal Figure 11

Rebuttal Figure 11:

hsp-16.2p::GFP induction in worms exposed to *E. coli* OP50 alone (naive), OP50 with 1-undecene odor and OP50 with heat shock.

Rebuttal Figure 12

Rebuttal Figure 12:

gst-4p::GFP induction in worms exposed to *E. coli* OP50 alone (naive), OP50 with 1-undecene odor and OP50 with 20 mM paraquat.

Other minor concerns that should be addressed:

Figure 1F: N2 appear to have 2 distinct populations, one responsive and one un-responsive. There is a similar separation in Figure S2D. What is the explanation?

RESPONSE: We and others have found such responses in behavioral assays. This may be linked to the expression of receptors for specific stimuli or epigenetic modification. We currently have no data to provide a solid explanation for this.

Welch's t-test is insensitive to equality of the variances, but does assume normal distribution. It should not be used for non-normal data (like Figure 1F).

RESPONSE: Thanks for your suggestion. We have carefully looked at our data and most of the data follow normal distribution. Therefore, we have applied two-tailed, unpaired *t*-test for analyses of choice index data presented in Figure 1F in the revised manuscript. This has been used by many other groups for the analysis of choice indices (Harris et al., 2014; Pereira et al., 2020; Worthy et al., 2018).

If the control data for Figure 4C is identical to the data in Figure 4A, this needs to be indicated.

RESPONSE: Thank you. We have provided separate controls across panels 4A, B, D and E for qRT-PCR in the revised Figure 4.

qRT-PCR is presented, for example, as "Relative mRNA expression N2 (1-Undecene / Naive)", but there are negative values, so this cannot be right. What is actually being represented?

RESPONSE: The negative values are arrived at by representing FC value less than 1 as $(-1/FC)$. This allows for better visualization of downregulation. For example, a fold change of 0.2 is a 5-fold downregulation but hard to visualize on positive axis for fold change. Thus, we use this method of representation (Dasgupta et al., 2020; Dixit et al., 2020).

The authors make a distinction between "younger" and "older" lawns. If 1-undecene is not produced by *P. aeruginosa* in liquid culture and/or at 37C, the differences observed would simply be a question of the time that the bacterial cultures have been in a situation compatible with 1-undecene synthesis. This could readily be checked by assaying *undA* expression under the different culture conditions.

RESPONSE: 1-Undecene is also produced in liquid culture (Rui et al., 2014; Timm et al., 2018). We have examined *undA* expression only on solid lawn where we study *C. elegans*-bacteria interaction and show behavioral changes. One could study the change in expression in liquid culture and change of media but that would not be relevant to our study. We also do not see an alteration in *undA* expression level in *rhlR* mutant (rebuttal Figure 8) suggesting that it may be independent of quorum sensing and population density.

References:

- Bargmann, C.I., Hartweg, E., and Horvitz, H.R. (1993). Odorant-selective genes and neurons mediate olfaction in *C. elegans*. *Cell* **74**, 515-527.
- Dasgupta, M., Shashikanth, M., Gupta, A., Sandhu, A., De, A., Javed, S., and Singh, V. (2020). NHR-49 transcription factor regulates immunometabolic response and survival of *Caenorhabditis elegans* during *Enterococcus faecalis* infection. *Infection and immunity* **88**.
- Dixit, A., Sandhu, A., Modi, S., Shashikanth, M., Koushika, S.P., Watts, J.L., and Singh, V. (2020). Neuronal control of lipid metabolism by STR-2 G protein-coupled receptor promotes longevity in *Caenorhabditis elegans*. *Aging cell* **19**, e13160.
- Dunbar, T.L., Yan, Z., Balla, K.M., Smelkinson, M.G., and Troemel, E.R. (2012). *C. elegans* detects pathogen-induced translational inhibition to activate immune signaling. *Cell host & microbe* **11**, 375-386.
- Estes, K.A., Dunbar, T.L., Powell, J.R., Ausubel, F.M., and Troemel, E.R. (2010). bZIP transcription factor zip-2 mediates an early response to *Pseudomonas aeruginosa* infection in *Caenorhabditis elegans*. *Proceedings of the National Academy of Sciences* **107**, 2153-2158.
- Ha, H.-i., Hendricks, M., Shen, Y., Gabel, C.V., Fang-Yen, C., Qin, Y., Colón-Ramos, D., Shen, K., Samuel, A.D., and Zhang, Y. (2010). Functional organization of a neural network for aversive olfactory learning in *Caenorhabditis elegans*. *Neuron* **68**, 1173-1186.
- Harris, G., Shen, Y., Ha, H., Donato, A., Wallis, S., Zhang, X., and Zhang, Y. (2014). Dissecting the signaling mechanisms underlying recognition and preference of food odors. *Journal of Neuroscience* **34**, 9389-9403.
- Hunziker, L., Bönisch, D., Groenhagen, U., Bailly, A., Schulz, S., and Weisskopf, L. (2015). *Pseudomonas* strains naturally associated with potato plants produce volatiles with high potential for inhibition of *Phytophthora infestans*. *Applied and environmental microbiology* **81**, 821-830.
- Jang, M.S., Toyoshima, Y., Tomioka, M., Kunitomo, H., and Iino, Y. (2019). Multiple sensory neurons mediate starvation-dependent aversive navigation in *Caenorhabditis elegans*. *Proceedings of the National Academy of Sciences* **116**, 18673-18683.
- Lo Cantore, P., Giorgio, A., and Iacobellis, N.S. (2015). Bioactivity of volatile organic compounds produced by *Pseudomonas tolaasii*. *Frontiers in microbiology* **6**, 1082.
- Meisel, J.D., Panda, O., Mahanti, P., Schroeder, F.C., and Kim, D.H. (2014). Chemosensation of bacterial secondary metabolites modulates neuroendocrine signaling and behavior of *C. elegans*. *Cell* **159**, 267-280.
- Pereira, A.G., Gracida, X., Kagias, K., and Zhang, Y. (2020). *C. elegans* aversive olfactory learning generates diverse intergenerational effects. *Journal of Neurogenetics*, 1-11.
- Reddy, K.C., Dror, T., Sowa, J.N., Panek, J., Chen, K., Lim, E.S., Wang, D., and Troemel, E.R. (2017). An intracellular pathogen response pathway promotes proteostasis in *C. elegans*. *Current Biology* **27**, 3544-3553. e3545.
- Reddy, K.C., Dror, T., Underwood, R.S., Osman, G.A., Elder, C.R., Desjardins, C.A., Cuomo, C.A., Barkoulas, M., and Troemel, E.R. (2019). Antagonistic paralogs control a switch between growth and pathogen resistance in *C. elegans*. *PLoS pathogens* **15**, e1007528.
- Rui, Z., Li, X., Zhu, X., Liu, J., Domigan, B., Barr, I., Cate, J.H., and Zhang, W. (2014). Microbial biosynthesis of medium-chain 1-alkenes by a nonheme iron oxidase. *Proceedings of the National Academy of Sciences* **111**, 18237-18242.
- Singh, J., and Aballay, A. (2019). Microbial colonization activates an immune fight-and-flight response via neuroendocrine signaling. *Developmental cell* **49**, 89-99. e84.
- Siripatrawan, U. (2008). Rapid differentiation between *E. coli* and *Salmonella typhimurium* using metal oxide sensors integrated with pattern recognition. *Sensors and Actuators B: Chemical* **133**, 414-419.
- Styer, K.L., Singh, V., Macosko, E., Steele, S.E., Bargmann, C.I., and Aballay, A. (2008). Innate immunity in *Caenorhabditis elegans* is regulated by neurons expressing NPR-1/GPCR. *Science* **322**, 460-464.

Timm, C.M., Lloyd, E.P., Egan, A., Mariner, R., and Karig, D. (2018). Direct growth of bacteria in headspace vials allows for screening of volatiles by gas chromatography mass spectrometry. *Frontiers in microbiology* 9, 491.

Troemel, E.R., Chu, S.W., Reinke, V., Lee, S.S., Ausubel, F.M., and Kim, D.H. (2006). p38 MAPK regulates expression of immune response genes and contributes to longevity in *C. elegans*. *PLoS Genet* 2, e183.

Worthy, S.E., Rojas, G.L., Taylor, C.J., and Glater, E.E. (2018). Identification of odor blend used by *Caenorhabditis elegans* for pathogen recognition. *Chemical senses* 43, 169-180.

Yoshida, K., Hirotsu, T., Tagawa, T., Oda, S., Wakabayashi, T., Iino, Y., and Ishihara, T. (2012). Odour concentration-dependent olfactory preference change in *C. elegans*. *Nature communications* 3, 1-11.

Zhu, J., Bean, H.D., Kuo, Y.-M., and Hill, J.E. (2010). Fast detection of volatile organic compounds from bacterial cultures by secondary electrospray ionization-mass spectrometry. *Journal of clinical microbiology* 48, 4426-4431.

-----END OF REBUTTAL-----

Dear Varsha,

Thank you for submitting your revised manuscript to The EMBO Journal. Your study has now been re-reviewed by the three referees and their comments are provided below.

As you can see, the referees appreciate the introduced revisions are overall supportive of the manuscript. The referees have a few more points that should be fairly straightforward to address. Regarding the point raised by referee #3 that the data doesn't sufficiently support that 1-undecene triggers a *P. aeruginosa*-specific response: the suggested experiment should be fairly straightforward to do or perhaps you have already done this? Let's discuss this further via video call or email.

When you submit your revised version will you also take care of the following points:

Please upload high resolution individual figures and remove the figures from the MS file. Place the figure legends at the end of the manuscript.

You can only have 5 keywords - you have at the moment 6.

Please check that there are figure callouts to Fig 4B+E and Fig S4G+H panels.

The supplementary file should be re-labelled as appendix. The figures and tables in the appendix needs to be referred to as 'Appendix Figure S#' and 'Appendix Table S#'. Please also correct callout in the text to the appendix figures/tables.

The movie legends should be removed from the appendix and ZIPed together with each movie file. The names and callouts in text needs to be corrected to 'Movie EV#'.

Methods needs correcting to Materials and Methods.

Appendix Fig S1 A panel label is missing.

Appendix Fig 3 has only one panel and so OK to refer simply as Figure 3.

We include a synopsis of the paper (see <http://emboj.embopress.org/>). Please provide me with a general summary statement and 3-5 bullet points that capture the key findings of the paper.

We also need a summary figure for the synopsis. The size should be 550 wide by [200-400] high (pixels). You can also use something from the figures if that is easier.

Our publisher has also done their pre-publication check on your manuscript. When you log into the manuscript submission system you will see the file "Data edited manuscript file". Please take a look at the word file and the comments regarding the figure legends and respond to the issues.

When you submit your revised manuscript please also include a point-by-point response also to the editorial points above.

That should be all - you can use the link below to upload the revised version.

With best wishes

Karin

Karin Dumstrei, PhD
Senior Editor
The EMBO Journal

Further information is available in our Guide For Authors:

The revision must be submitted online within 90 days; please click on the link below to submit the revision online before 23rd Jun 2021.

Referee #1:

In this revised manuscript, the authors more convincingly show that 1-undecene serves as a volatile *Pseudomonas*-specific pathogen associated molecular pattern. This is a significant finding of general interest. In my opinion, the authors have thoroughly addressed my concerns and I now consider this manuscript suitable for publication.

There do remain some minor proofreading or textual issues, some of which are listed below:

Calcium imaging figures do not indicate *n* or the number of imaging sessions.

Figure 3E - does not indicate under what imaging condition this representative image was taken.

Figure 4F - should indicate whether worms were exposed on partial or full lawns.

The statement (line 219), "We did not observe calcium response to 1-undecene in AWA, AWCon or AWCoFF neurons (Fig. S3A) - There appear to be small responses in AWC, so would it not perhaps be more accurate to state that you did not observe large responses in AWC?"

Referee #2:

The authors are responsive to my comments and addressed the concerns in the revised manuscript.

Referee #3:

The manuscript is much improved. I do still, however, have reservations concerning one of the authors' central claims, exemplified by the sentence in the Discussion, "The most relevant evidence for 1-undecene as a pathogen-associated molecular pattern comes from the fact that exposure of worms to volatile alone induces upregulation of immune response genes specific to *P. aeruginosa*".

In their rebuttal, the authors write, "irg-1, irg-2 and irg-3 are specific to *P. aeruginosa* as shown previously by Troemel lab (Estes et al., 2010)" and "Troemel lab has shown that irg-1, irg-2 and irg-3 are specific to *P. aeruginosa* infection and translational inhibition by the pathogen (Dunbar et al., 2012; Estes et al., 2010)".

Yet, as I pointed out, irg-2 expression is induced by Orsay virus, and irg-1 and irg-2 expression is strongly induced by cycloheximide. Therefore, their induction is not "specific to *P. aeruginosa* infection and translational inhibition by the pathogen", but to translational inhibition more generally. I would thus have expected the authors to have looked, for example, at atf-4 expression after 1-undecene exposure (by qRT-PCR).

These reservations are also due to the selective manner in which the authors interpret their data. They report a 2-fold increase for irg-1 expression as an induction (Fig. 4A), yet while the expression of both fmo-2 and acs-2 increased by more than 2-fold wrote, "We found that 1-undecene odor exposure did not induce expression of transcripts of fmo-2, acs-2, lipl-1, lipl-3, cpr-4, cpr-5, asp-14 and lys-3 (Fig. S4D)". They can't have it both ways.

Further they write, "We also analyzed the induction of heat shock response or oxidative stress response using hsp-16.2p::GFP and gst-4p::GFP respectively (Link et al., 1999; Link and Johnson,

2002). We found that 1-undecene odor exposure did not disrupt cellular homeostasis". These results (for which the reader should be referred to FigS4G,H) are not entirely convincing, in part as the positive controls give such weak signals. Their own results for *fmo-2* show how qRT-PCR is a more sensitive assay for changes in gene expression than a GFP reporter. Is there a reason why a more sensitive, qRT-PCR analysis was not performed?

As a very minimum, the authors need to tone down substantially their claim that 1-undecene is triggering a *P. aeruginosa*-specific response, but ideally conduct the experiments to define whether or not 1-undecene has more general effects on host physiology, by measuring for example, *atf-4* mRNA levels before and after exposure.

Minor points

"qRT-PCR is presented, for example, as "Relative mRNA expression N2 (1-Undecene / Naive)", but there are negative values, so this cannot be right. What is actually being represented?
RESPONSE: The negative values are arrived at by representing FC value less than 1 as $(-1/FC)$. This allows for better visualization of downregulation. For example, a fold change of 0.2 is a 5-fold downregulation but hard to visualize on positive axis for fold change".

That's fine, but this information must be included in the figure legends!

When I wrote, "There are some recent studies that should have been cited, including Ringstad's on O₂/CO₂ sensing", I was referring to "Toll-like Receptor Signaling Promotes Development and Function of Sensory Neurons Required for a *C. elegans* Pathogen-Avoidance Behavior". Brandt JP, Ringstad N. *Curr Biol*. 2015. This needs to be included at, "except TOL-1 which has a limited role in *C. elegans* microbe interactions (Pradel et al., 2007; Tenor and Aballay, 2008)".

Referee #1:

In this revised manuscript, the authors more convincingly show that 1-undecene serves as a volatile *Pseudomonas*-specific pathogen associated molecular pattern. This is a significant finding of general interest. In my opinion, the authors have thoroughly addressed my concerns and I now consider this manuscript suitable for publication.

RESPONSE: Thank you.

There do remain some minor proofreading or textual issues, some of which are listed below:

Calcium imaging figures do not indicate n or the number of imaging sessions.

RESPONSE: Number of animals imaged in a single session are included the legend for Figure 3. Two to three imaging sessions were performed for each strain.

Figure 3E - does not indicate under what imaging condition this representative image was taken.

RESPONSE: 3E represents GCaMP3 fluorescence in one of the AWB neurons 129 s after 1-undecene exposure and after 1.2 seconds (131.2 seconds on x-axis) of 1-undecene withdrawal. This information is available in the legend of Figure 3.

Figure 4F - should indicate whether worms were exposed on partial or full lawns.

RESPONSE: The worms were exposed on full lawns of *P. aeruginosa*.

The statement (line 219), "We did not observe calcium response to 1-undecene in AWA, AWCon or AWCo^{ff} neurons (Fig. S3A) - There appear to be small responses in AWC, so would it not perhaps be more accurate to state that you did not observe large responses in AWC?"

RESPONSE: We have modified our statement to '*We observed none or very small calcium response to 1-undecene withdrawal in AWA, AWC^{on} or AWC^{off} neurons*'.

Referee #2:

The authors are responsive to my comments and addressed the concerns in the revised manuscript.

RESPONSE: Thank you.

Referee #3:

The manuscript is much improved. I do still, however, have reservations concerning one of the authors' central claims, exemplified by the sentence in the Discussion, "The most relevant evidence for 1-undecene as a pathogen-associated molecular pattern comes from the fact that exposure of worms to volatile alone induces upregulation of immune response genes specific to *P. aeruginosa*".

In their rebuttal, the authors write, "irg-1, irg-2 and irg-3 are specific to *P. aeruginosa* as shown previously by Troemel lab (Estes et al., 2010)" and "Troemel lab has shown that irg-1, irg-2 and irg-3 are specific to *P. aeruginosa* infection and translational inhibition by the pathogen (Dunbar et al., 2012; Estes et al., 2010)".

Yet, as I pointed out, irg-2 expression is induced by Orsay virus, and irg-1 and irg-2 expression is strongly induced by cycloheximide. Therefore, their induction is not "specific to *P. aeruginosa* infection and translational inhibition by the pathogen", but to translational inhibition more generally. I would thus have expected the authors to have looked, for example, at *atf-4* expression after 1-undecene exposure (by qRT-PCR).

RESPONSE: Thanks for your suggestions. We have performed *atf-4* (Glover-Cutter et al., 2013) qRT-PCR in worms exposed to 1-undecene odor. As shown in the rebuttal figure 1, we see just about 1.3-fold increase in *atf-4* transcript in 1-undecene exposed worms compared to the naïve worms. We also examined an endoplasmic reticulum chaperone, *hsp-4*, (Shen et al., 2001) and found no significant change due to 1-undecene exposure. We have included this data in Appendix Figure S4I in the revised manuscript.

Rebuttal Figure 1

Rebuttal Figure 1:

Real time PCR analysis of transcripts for *atf-4* and *hsp-4* in N2 worms exposed to 1-undecene odor for 2 hours over naive N2 worms. n = 3. Error bars indicate SEM.

These reservations are also due to the selective manner in which the authors interpret their data. They report a 2-fold increase for *irg-1* expression as an induction (Fig. 4A), yet while the expression of both *fmo-2* and *acs-2* increased by more than 2-fold wrote, "We found that 1-undecene odor exposure did not induce expression of transcripts of *fmo-2*, *acs-2*, *lipl-1*, *lipl-3*, *cpr-4*, *cpr-5*, *asp-14* and *lys-3* (Fig. S4D)". They can't have it both ways.

RESPONSE: Thanks for your comments. We have modified our statement in the revised manuscript to 'We found that 1-undecene odor exposure showed little or no induction in expression of transcripts of *fmo-2*, *acs-2*, *lipl-1*, *lipl-3*, *cpr-4*, *cpr-5*, *asp-14* and *lys-3*'. Both *fmo-2* and *acs-2* are induced tens to hundreds of folds during infection of worms with *E. faecalis* or pathogenic yeast *Cryptococcus neoformans* (Dasgupta et al., 2020) while we see 2-3-fold increase in 1-undecene exposed worms. The 2-fold change in exposed over naive worms is rather weak in comparison to induction seen with pathogenic Gram-positive bacteria or yeast.

Further they write, "We also analyzed the induction of heat shock response or oxidative stress response using *hsp-16.2p::GFP* and *gst-4p::GFP* respectively (Link et al., 1999; Link and Johnson, 2002). We found that 1-undecene odor exposure did not disrupt cellular homeostasis". These results (for which the reader should be referred to FigS4G, H) are not entirely convincing, in part as the positive controls give such weak signals. Their own results for *fmo-2* show how qRT-PCR is a more sensitive assay for changes in gene expression than a GFP reporter. Is there a reason why a more sensitive, qRT-PCR analysis was not performed?

RESPONSE: Thank you for your suggestions. We have analyzed transcript levels of *hsp16.2* and *gst-4* by qRT-PCR. As shown in the rebuttal figure 2, we see no significant change in the transcript levels for *hsp16.2* and two-fold increase for *gst-4*.

Rebuttal Figure 2

Rebuttal Figure 2:

Real time PCR analysis of transcripts for *hsp-16.2* and *gst-4* in N2 worms exposed to 1-undecene for 2 hours over naive N2 worms. n = 3. * P ≤ 0.05 as determined by two-tailed unpaired t-test with Welch's correction. Error bars indicate SEM.

As a very minimum, the authors need to tone down substantially their claim that 1-undecene is triggering a *P. aeruginosa*-specific response, but ideally conduct the experiments to define whether or not 1-undecene has more general effects on host physiology, by measuring for example, *atf-4* mRNA levels before and after exposure.

RESPONSE: qRT-PCR analyses of 8 immune effectors specific to pathogenic yeast and Gram-positive bacteria (Appendix Fig. S4D) indicated that they are either not induced at all or induced to about two folds for *fmo-2* and *acs-2*. This is miniscule in comparison to 100-fold or higher induction seen for *fmo-2* in response to infection with pathogenic bacteria (Dasgupta 2020). In addition, we find less than two-fold change in levels of transcripts for *atf-4*, *hsp-4* and *hsp16.2*, and small increase in *gst-4* transcript (Rebuttal Figs. 1 and 2). Microscopy reveals no increase in reporter expression for either *hsp16.2* or *gst-4* (Fig. S4G and S4H).

Minor points

"qRT-PCR is presented, for example, as "Relative mRNA expression N2 (1-Undecene / Naive)", but there are negative values, so this cannot be right. What is actually being represented?

RESPONSE: The negative values are arrived at by representing FC value less than 1 as (-1/FC). This allows for better visualization of downregulation. For example, a fold change of 0.2 is a 5-fold downregulation but hard to visualize on positive axis for fold change". That's fine, but this information must be included in the figure legends!

RESPONSE: We have included this in the legend of figure 4 in the revised manuscript.

When I wrote, "There are some recent studies that should have been cited, including Ringstad's on O₂/CO₂ sensing", I was referring to "Toll-like Receptor Signaling Promotes Development and Function of Sensory Neurons Required for a *C. elegans* Pathogen-Avoidance Behavior". Brandt JP, Ringstad N. *Curr Biol*. 2015. This needs to be included at, "except TOL-1 which has a limited role in *C. elegans* microbe interactions (Pradel et al., 2007; Tenor and Aballay, 2008)".

RESPONSE: We have included the in the revised manuscript.

REFERENCES:

Dasgupta, M., Shashikanth, M., Gupta, A., Sandhu, A., De, A., Javed, S., and Singh, V. (2020). NHR-49 transcription factor regulates immunometabolic response and survival of *Caenorhabditis elegans* during *Enterococcus faecalis* infection. *Infection and immunity* 88.

Glover-Cutter, K.M., Lin, S., and Blackwell, T.K. (2013). Integration of the unfolded protein and oxidative stress responses through SKN-1/Nrf. *PLoS Genet* 9, e1003701.

Shen, X., Ellis, R.E., Lee, K., Liu, C.-Y., Yang, K., Solomon, A., Yoshida, H., Morimoto, R., Kurnit, D.M., and Mori, K. (2001). Complementary signaling pathways regulate the unfolded protein response and are required for *C. elegans* development. *Cell* 107, 893-903.

EDITORIAL COMMENTS

Please upload high resolution individual figures and remove the figures from the MS file. Place the figure legends at the end of the manuscript.

RESPONSE: Done.

You can only have 5 keywords - you have at the moment 5.

RESPONSE: We have included 5 keywords.

Please check that there are figure callouts to Fig 4B+E and Fig S4G+H panels.

RESPONSE: Call outs to Fig 4B+E and Fig S4G+H panels have been included in the revised manuscript.

The supplementary file should be re-labelled as appendix. The figures and tables in the appendix need to be referred to as 'Appendix Figure S#' and 'Appendix Table S#'. Please also correct callout in the text to the appendix figures/tables.

RESPONSE: We have labelled supplementary files as appendix. We have corrected callouts throughout the revised manuscript.

The movie legends should be removed from the appendix and ZIPed together with each movie file. The names and callouts in text needs to be corrected to 'Movie EV#'.

RESPONSE: Movie legends have been ZIPed together with each movie. The names and callouts have been corrected in the text.

Methods needs correcting to Materials and Methods.

RESPONSE: Corrected in the revised manuscript.

Appendix Fig S1 A panel label is missing.

RESPONSE: Corrected.

Appendix Fig 3 has only one panel and so OK to refer simply as Figure 3.

RESPONSE: Corrected.

We include a synopsis of the paper (see <http://emboj.embopress.org/>). Please provide me with a general summary statement and 3-5 bullet points that capture the key findings of the paper.

RESPONSE:

Summary Statement: Well-studied pathogen-associated molecular patterns (PAMPs) are constituents of microbial cells such as certain components of their cell wall. Here, we show that a volatile molecule produced by *Pseudomonas aeruginosa* activates a pathogen-specific immune response in *Caenorhabditis elegans* host. This study presents a new paradigm for pathogen recognition resulting in activation of flight or fight response.

- 1-undecene volatile produced by *P. aeruginosa* activates flight response in *C. elegans* on short time scale of second to minutes.
- Longer exposure of worms to 1-undecene induces a pathogen-specific immune response.
- Both flight and fight response induced by 1-undecene rely on the AWB olfactory neurons of worms.
- Preexposure of worms to 1-undecene protects them against a subsequent infection with *P. aeruginosa*.
- 1-Undecene serves as a pathogen-associated molecular pattern (PAMP) for *C. elegans*.

We also need a summary figure for the synopsis. The size should be 550 wide by [200-400] high (pixels). You can also use something from the figures if that is easier.

RESPONSE: We have uploaded a summary figure.

Dear Varsha,

Thank you for submitting your revised manuscript. I have now had a chance to take a look at it and I appreciate the introduced changes.

I am therefore very pleased to accept the manuscript for publication here.

Congratulations on a nice study

Best Karin

Karin Dumstrei, PhD
Senior Editor
The EMBO Journal

Please note that it is EMBO Journal policy for the transcript of the editorial process (containing referee reports and your response letter) to be published as an online supplement to each paper. If you do NOT want this, you will need to inform the Editorial Office via email immediately. More information is available here: https://emboj.embopress.org/about#Transparent_Process

Your manuscript will be processed for publication in the journal by EMBO Press. Manuscripts in the PDF and electronic editions of The EMBO Journal will be copy edited, and you will be provided with page proofs prior to publication. Please note that supplementary information is not included in the proofs.

Should you be planning a Press Release on your article, please get in contact with embojournal@wiley.com as early as possible, in order to coordinate publication and release dates.

If you have any questions, please do not hesitate to call or email the Editorial Office. Thank you for your contribution to The EMBO Journal.

Corresponding Author Name: Varsha Singh

Manuscript Number: EMBOJ-2020-106938